# Superclass-Conditional Gaussian Mixture Model For Learning Fine-Grained Embeddings

**Jingchao Ni**[1]**, Wei Cheng**[1]**, Zhengzhang Chen**[1]**, Takayoshi Asakura**[2]**,
Tomoya Soma**[2]**, Sho Kato**[3]**, Haifeng Chen**[1]
[1]NEC Laboratories America, [2]NEC Corporation, [3]Renascience, Inc.
[1]{jni,weicheng,zchen,haifeng}@nec-labs.com,
[2]{takayoshi.asakura,tomoya-s}@nec.com, [3]kato@renascience.co.jp

## ABSTRACT

Learning fine-grained embeddings is essential for extending the generalizability of models pre-trained on "coarse" labels (*e.g.*, animals). It is crucial to fields for which fine-grained labeling (*e.g.*, breeds of animals) is expensive, but fine-grained prediction is desirable, such as medicine. The dilemma necessitates adaptation of a "coarsely" pre-trained model to new tasks with a few "finer-grained" training labels. However, coarsely supervised pre-training tends to suppress intra-class variation, which is vital for cross-granularity adaptation. In this paper, we develop a training framework underlain by a novel superclass-conditional Gaussian mixture model (SCGM). SCGM imitates the generative process of samples from hierarchies of classes through latent variable modeling of the fine-grained subclasses. The framework is agnostic to the encoders and only adds a few distribution related parameters, thus is efficient, and flexible to different domains. The model parameters are learned end-to-end by maximum-likelihood estimation via a principled Expectation-Maximization algorithm. Extensive experiments on benchmark datasets and a real-life medical dataset indicate the effectiveness of our method.

## 1 INTRODUCTION

Training deep models with sufficient generalizability is of fundamental importance, which demands immense training data with fine-grained annotations (Krizhevsky et al., 2012; Brown et al., 2020). In many fields, however, data labeling requires domain-specific knowledge, such as medicine (Sohoni et al., 2020), thus is prohibitive, and infeasible to be exhaustive. In this case, data for model training may only be "coarsely" labeled, while later the model is tested on a finer-grained classification task (Bukchin et al., 2021). For example, consider an event prediction task for dialysis patients (Inaguma et al., 2019). Hemodialysis is a major renal replacement therapy for patients with end-stage renal failure. These patients have to take hemodialysis thrice a week, each lasts for 4-5 hours. During the treatment, unexpected events, such as muscle cramp, perspiration, and dizziness, may happen as a result of lowering blood pressure, which need intensive medical care, thus should always be avoided. It is therefore an important medical issue to predict such events before an initiation of hemodialysis. In this task, binary labels, which mark the incidence of an event, can be collected. In contrast, finer-grained labels that annotate different subtypes of events are seldom recorded. Since distinguishing different subtypes facilitates precise diagnoses, and helps physicians assess the risk for deciding whether to perform a hemodialysis (with certain precautions), it is desirable that a model trained with coarse (binary) labels can perform well on a finer-grained multi-class (subtypes) task.

To fill the gap of granularity between the training and testing scenarios, a practical way is to collect a few new records for a patient, with their fine-grained annotations. These data constitute a *support set* for fine-tuning a pre-trained model to the specific data distribution induced by the annotations of the target patient, for whom the adapted model is used for future predictions. Although massive fine-grained annotation is impractical, annotating a few-shot set is feasible. In this work, we are interested in such a Cross-Granularity Few-Shot (CGFS) learning problem, where a model pre-trained on a set of coarse classes (denoted as *superclasses*), needs to adapt to an unseen set of fine-grained target classes (denoted as *subclasses*). The target subclasses could be descendants of the superclasses (as in

the aforementioned example), or could descend from other superclasses that are unobserved during pre-training. To be practical, the adaptation should only use a few samples from the subclasses.

The CGFS problem is not limited to the above application. It occurs in a model's lifespan when an application requires separating some subclasses from the superclasses yet when the training dataset was created these subclasses were unannotated. For example, it could occur in detecting rare pathology or variants using medical images that were coarsely described (Oakden-Rayner et al., 2020), or personalizing a generic model trained on all historical users to a specific customer (Luo et al., 2020).

Despite its significance, CGFS cannot be trivially solved by regularly training models with coarse labels, because typical losses for supervised learning aim to maximize inter-class boundaries but neglect intra-class variation. Thus, subclasses may arbitrarily and unevenly spread in every superclass. Recently, Bukchin et al. (2021) proposed to integrate coarse supervision and contrastive learning (within superclasses) for solving CGFS. However, their approach cannot be readily used for medical records which typically contain static profiles and time series (Che et al., 2016). This is due to the absence of a standard augmentation method for generating constrastive pairs on some data other than images. Also, since contrastive learning does not model subclass explicitly, their solution could be suboptimal (as evaluated in Sec. 4). Moreover, as their model was built upon MoCo (He et al., 2020), and maintains many dictionaries for data sampling, its computational costs are high (Sec. 4).

In this work, we propose a novel Superclass-Conditional Gaussian Mixture model (SCGM) to learn fine-grained embeddings for the CGFS problem. Our contributions are summarized as follows.

- SCGM is agnostic to the encoder, thus is flexible to different applications. It models the generation of samples from hierarchical classes, and explicitly represents the unobserved subclasses by latent variables, without assuming their identities. It dynamically computes a Gaussian mixture for every sample conditioned on its superclass, and the model forms a hierarchy of Gaussian mixtures.

- SCGM only adds a small overhead to an encoder, for parameterizing its distributions, thus is efficient. The model parameters are learned end-to-end by maximum likelihood estimation via a principled Expectation-Maximization (EM) algorithm. We also theoretically linked our loss function to InfoNCE (Oord et al., 2018), explaining its effectiveness from a contrastive perspective.

- In the experiments, we evaluated SCGM on both benchmark image datasets and a real-life medical dataset. Since SCGM is compatible with contrastive learning, we also tested it with a momentum encoder (He et al., 2020). The results demonstrate SCGM on generic encoders has already outperformed the state-of-the-art (SOTA) baselines, with less computational costs, and it achieves boosted performance in some cases with momentum contrast integrated.

## 2 RELATED WORK

To our best knowledge, this is the first work to develop an SCGM model for underlying a framework that enables tackling CGFS across domains. The most relevant work (Bukchin et al., 2021) combines superclass-wise contrastive learning with coarse classification for preserving intra-class variation. Another work (Yang et al., 2021) used a three-step approach that pseudo-labels the embeddings (pretrained by coarse classification and batch-wise contrastive learning) by clustering every superclass. The pseudo-fine-labels were used to re-train the encoder. A similar three-step method (Sohoni et al., 2020) used a different loss for maximizing the worst-case expected accuracy over the pseudo-labeled subclasses. As discussed before, the former two methods require non-trivial searches for a suitable data augmentation method, which may be unavailable for some non-image data. By splitting training steps, the latter two methods could lead to suboptimal pseudo-labeling, and misleading labels could confuse the downstream steps. In contrast, our model is end-to-end. It explicitly infers the posterior of the subclass during coarse training. Moreover, it can achieve better performance without using as many computational resources as the model of (Bukchin et al., 2021) (as evaluated in Sec. 4).

**Learning with coarse supervision.** Several works deal with coarse/fine labels from the perspective of weakly supervised learning (Zhou, 2018) than to tackle the CGFS problem, including training methods that take advantage of a mixture of either balanced (Ristin et al., 2015; Guo et al., 2018; Taherkhani et al., 2019) or unbalanced (Hsieh et al., 2019; Liu et al., 2019; Robinson et al., 2020) coarse and fine labels. Among them, Liu et al. (2019) addressed a few-shot learning problem, but the model training assumes access to some fine labels and a graph of class hierarchy, which are unavailable in our problem. Thus it cannot be adapted to solve our problem.

**Few-shot learning.** Meta-learning has become a popular idea to handle the few-shot learning problem, which derives metric-based (Vinyals et al., 2016; Snell et al., 2017) and optimization-based methods (Finn et al., 2017; Nichol et al., 2018). The idea has been extended to semi-supervised (Ren et al., 2018), unsupervised (Hsu et al., 2018) and semantics-augmented (Xing et al., 2019) scenarios, when labels are scarce. However, none of them explores coarse labels for cross-granularity learning. Recently, many works observed that learning embeddings on all classes (without episodic training), followed by simple fine-tuning, is superior to SOTA meta-learning methods (Wang et al., 2019; Dhillon et al., 2020; Tian et al., 2020). In this work, similar to (Bukchin et al., 2021), we focus on this paradigm to learn useful embeddings, and do not use meta-learning for pre-training.

**Embedding methods.** The emerging self-supervised methods, such as contrastive methods (Oord et al., 2018; Chen et al., 2020a; He et al., 2020), are appealing in their ability to attain comparable embeddings to the supervised counterparts, and even surpass them when transferring to other tasks (He et al., 2020; Tian et al., 2020). Beyond instance-wise contrast, recent methods have explored instance-cluster contrast to boost performance (Caron et al., 2020; Li et al., 2020). In unsupervised methods, joint embedding and clustering has been found beneficial (Caron et al., 2018; Asano et al., 2020). Whereas, these methods never exploited coarse supervision. Thus, their embeddings/clusters do not necessarily reflect intra-class variation, which is important to the CGFS task.

Vanilla Gaussian mixture (GM) model is unfit for the CGFS scenario. A recent work (Manduchi et al., 2021) extended GM to constrained clustering by conditioning every sample with a prior clustering preference. It is neither supervised by coarse classes nor hierarchically structured. Conventional hierarchical GM is used for hierarchical clustering (Goldberger & Roweis, 2005; Olech & Paradowski, 2016; Athey et al., 2019) by applying GM agglomeratively or divisively. These unsupervised methods only infer clusters, but do not pre-train embedding models for task adaptation.

## 3 SUPERCLASS-CONDITIONAL GAUSSIAN MIXTURE MODEL

Firstly, a word about some notations. Let $\mathcal{D}_{\text{train}} = \{(\mathbf{x}_i, y_i)\}_{i=1}^n$ be $n$ sample-label training pairs, where $y_i \in \mathcal{Y}_{\text{super}} = \{1, ..., c\}$ is a *superclass* label. Each $\mathbf{x}_i$ is associated with a latent (unobserved) *subclass* label $\hat{y}_i \in \mathcal{Y}_{\text{sub}} = \{1, ..., s\}$. $\mathcal{Y}_{\text{sub}}$ relates to $\mathcal{Y}_{\text{super}}$ by a hierarchical structure, *i.e.*, $\mathcal{Y}_{\text{sub}}$ can be partitioned into $c$ disjoint sets $\mathcal{Y}_{\text{sub-1}}, ..., \mathcal{Y}_{\text{sub-}c}$, such that if $\hat{y}_i \in \mathcal{Y}_{\text{sub-}j}$, then $y_i = j$ ($1 \leq j \leq c$). Let $f_{\boldsymbol{\theta}}$ be an encoder (*i.e.*, backbone network) that is trained on $\mathcal{D}_{\text{train}}$ (without knowing $\mathcal{Y}_{\text{sub}}$). It maps $\mathbf{x}_i$ to a $d$-dimensional feature $f_{\boldsymbol{\theta}}(\mathbf{x}_i)$. At test time, given a $k$-shot support set for a subset $\mathcal{Y}_{\text{sub}}^m \subseteq \mathcal{Y}_{\text{sub}}$ of $m$ subclasses, *i.e.*, $\mathcal{D}_{\text{support}} = \{(\mathbf{x}_i, \hat{y}_i) | \hat{y}_i \in \mathcal{Y}_{\text{sub}}^m\}_{i=1}^{mk}$, the task is to train a classifier $C: \mathbb{R}^d \to \mathcal{Y}_{\text{sub}}^m$ with optimal accuracy on a test set of $\mathcal{Y}_{\text{sub}}^m$ subclasses. In our experiments, we also explored the case when the subclasses belong to superclasses that are not used for pre-training. As discussed in Sec. 2, our focus is to train $f_{\boldsymbol{\theta}}$ for good embeddings without modifying $f_{\boldsymbol{\theta}}$ during the adaptation–a paradigm with SOTA few-shot performance (Dhillon et al., 2020; Tian et al., 2020).

Formally, let $f_{\boldsymbol{\theta}}(\mathbf{x}_i) = \mathbf{v}_i$, our goal is to find the model parameter $\boldsymbol{\theta}$ that maximizes the likelihood of the posterior distribution $p_{\boldsymbol{\theta}}(y_i|\mathbf{v}_i)$ on the observed data in $\mathcal{D}_{\text{train}}$ for classification tasks (Grathwohl et al., 2019). To model the unobserved subclasses, we associate every $\mathbf{v}_i$ with a latent variable $z_i$ to indicate to which subclass $\mathbf{v}_i$ belongs. Suppose there are $r$ possible subclasses, the log-likelihood to maximize can be rewritten by marginalizing out the latent variables.

$$\ell(\mathcal{D}_{\text{train}}; \boldsymbol{\theta}) = \frac{1}{n} \sum_{i=1}^n \log\left[p_{\boldsymbol{\theta}}(y_i|\mathbf{v}_i)\right] = \frac{1}{n} \sum_{i=1}^n \log\left[\sum_{z_i=1}^r p(y_i|z_i)p_{\boldsymbol{\theta}}(z_i|\mathbf{v}_i)\right] \tag{1}$$

where the distribution $p_{\boldsymbol{\theta}}(z_i|\mathbf{v}_i)$ specifies the subclass membership of $\mathbf{v}_i$, and $p(y_i|z_i)$ associates $z_i$ with a subclass partition $\mathcal{Y}_{\text{sub-}y_i}$. Unlike some previous works (Sohoni et al., 2020), which searched the number of subclasses for every superclass using a quality metric, we only assume a total number of $r$ subclasses, and seek to infer their relationship with superclasses, *i.e.*, $p(y_i|z_i)$, without any prior on subclass partitions, so that the model is more generalizable.

### 3.1 PARAMETERIZATION OF PROBABILITIES

In Eq. (1), both $p(y_i|z_i)$ and $p_{\boldsymbol{\theta}}(z_i|\mathbf{v}_i)$ are not specified, and involve non-parametric variables. To make it solvable, we introduce our SCGM that models the generative process of the embeddings.

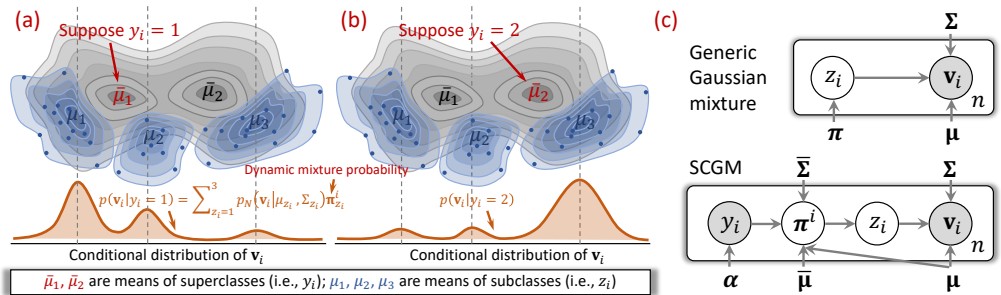

Figure 1: An illustration of (a) embedding distribution tuned by SCGM via dynamic mixture probabilities $\boldsymbol{\pi}^i$, when $y_i = 1$, and (b) when $y_i = 2$, where (c) summarizes SCGM's graphical model.

Let $\boldsymbol{\mu}_j$ and $\boldsymbol{\Sigma}_j$ ($1 \leq j \leq r$) be the mean and variance of the $j$-th mixture component, and $\pi_j$ be its corresponding mixture probability. In a generic GM model, the generation of $\mathbf{v}_i$ involves two steps: (1) draw a latent variable $z_i$ from a categorical distribution on all mixture components, and (2) draw $\mathbf{v}_i$ from the Gaussian distribution $\mathcal{N}(\boldsymbol{\mu}_{z_i}, \boldsymbol{\Sigma}_{z_i})$ (Bishop, 2006), as illustrated by Fig. 1(c).

In step (1), the categorical distribution is usually defined on $\boldsymbol{\pi} = [\pi_1, ..., \pi_r] \in \mathbb{R}^r$, *i.e.*, the static mixture probabilities, which cannot reflect different samples' superclasses. In light of this, for each $\mathbf{v}_i$, we propose to define $\boldsymbol{\pi}^i = [\pi_1^i, ..., \pi_r^i]$ to be conditional on $y_i$, which dynamically changes w.r.t. different samples. To this end, we introduce $\bar{\boldsymbol{\mu}}_{y_i}$ and $\bar{\boldsymbol{\Sigma}}_{y_i}$ as the mean and variance of superclass $y_i$, and compute $\pi_j^i = \text{softmax}(-D_{y_i}(\boldsymbol{\mu}_j)^2/2)$, where $D_{y_i}(\cdot)$ measures the Mahalanobis distance between a data point and the superclass distribution $\mathcal{N}(\bar{\boldsymbol{\mu}}_{y_i}, \bar{\boldsymbol{\Sigma}}_{y_i})$. Basically, $\pi_j^i$ evaluates the density ratio of the mixture component mean $\boldsymbol{\mu}_j$, by regarding it as a sample from $y_i$'s distribution.

$$\pi_j^i = \frac{\exp\left(-\frac{1}{2}(\boldsymbol{\mu}_j - \bar{\boldsymbol{\mu}}_{y_i})^\top \bar{\boldsymbol{\Sigma}}_{y_i}^{-1}(\boldsymbol{\mu}_j - \bar{\boldsymbol{\mu}}_{y_i})\right)}{\sum_{j'=1}^r \exp\left(-\frac{1}{2}(\boldsymbol{\mu}_{j'} - \bar{\boldsymbol{\mu}}_{y_i})^\top \bar{\boldsymbol{\Sigma}}_{y_i}^{-1}(\boldsymbol{\mu}_{j'} - \bar{\boldsymbol{\mu}}_{y_i})\right)} = \frac{p_N(\boldsymbol{\mu}_j | \bar{\boldsymbol{\mu}}_{y_i}, \bar{\boldsymbol{\Sigma}}_{y_i})}{\sum_{j'=1}^r p_N(\boldsymbol{\mu}_{j'} | \bar{\boldsymbol{\mu}}_{y_i}, \bar{\boldsymbol{\Sigma}}_{y_i})} \tag{2}$$

where $p_N(\cdot|\cdot)$ represents the density function of a multivariate Gaussian distribution.

Our generative process can be summarized as

1. for each sample index $i$:
   (a) draw a superclass label $y_i \sim \text{Categorical}(p(y_i))$
   (b) for $j = 1, ..., r$:
      i. compute $\pi_j^i = \text{softmax}(-D_{y_i}(\boldsymbol{\mu}_j)^2/2)$ using Mahalanobis distance $D_{y_i}(\cdot)$
   (c) draw a latent subclass variable for the $i$-th sample $z_i \sim \text{Categorical}([\pi_1^i, ..., \pi_r^i])$
   (d) draw a feature vector $\mathbf{v}_i \sim \mathcal{N}(\boldsymbol{\mu}_{z_i}, \boldsymbol{\Sigma}_{z_i})$

where $p(y_i)$ is a prior on superclasses, which can be drawn from a Dirichlet distribution $\text{Dir}(\boldsymbol{\alpha})$ (the conjugate of categorical distribution), and $\boldsymbol{\alpha}$ can be estimated by the ratios of different labels in $\mathcal{D}_{\text{train}}$. $p(y_i)$ can also be set as a uniform prior, for unknown datasets and better generalization.

As can be seen, steps (a)(b) specify a (static) mixture of superclasses, and steps (c)(d) specify a (dynamic) mixture of subclasses. The whole process establishes a hierarchical GM distribution. Fig. 1(a)(b) illustrate the process using 2 superclasses and 3 subclasses, where the embedding distribution $p(\mathbf{v}_i|y_i)$ dynamically changes w.r.t. $\mathbf{v}_i$'s superclass as a result of updated $\boldsymbol{\pi}^i$. For example, when $y_i = 1$ (Fig. 1(a)), $\mathbf{v}_i$ is more likely to be generated from the first two subclasses. This $y_i$-adjusted Gaussian mixture distribution highlights the first two subclasses, and will be used to fit data with $y_i = 1$ during model training. Fig. 1(c) bottom illustrates the graphical model of SCGM, which distinguishes itself from the generic GM in Fig. 1(c) by (1) the condition $y_i$, and (2) the variable $\boldsymbol{\pi}^i$.

Now, if we consider $\bar{\boldsymbol{\mu}}_{y_i}$ and $\boldsymbol{\mu}_{z_i}$ as the *surrogate representations* of $y_i$ and $z_i$ in the embedding space, we can specify two probabilities $p(z_i|y_i)$ and $p(\mathbf{v}_i|z_i)$ using their corresponding densities:

$$p(z_i|y_i) = p_N(\boldsymbol{\mu}_{z_i}|\bar{\boldsymbol{\mu}}_{y_i}, \bar{\boldsymbol{\Sigma}}_{y_i}), \qquad p(\mathbf{v}_i|z_i) = p_N(\mathbf{v}_i|\boldsymbol{\mu}_{z_i}, \boldsymbol{\Sigma}_{z_i}) \tag{3}$$

from which, the probabilities $p(y_i|z_i)$ and $p(z_i|\mathbf{v}_i)$ in Eq. (1) can be derived by using Bayesian rules as (the detailed derivation is deferred to Appendix A.1)

$$p_\phi(y_i|z_i) = \frac{p_N(\boldsymbol{\mu}_{z_i}|\bar{\boldsymbol{\mu}}_{y_i}, \bar{\boldsymbol{\Sigma}}_{y_i})p(y_i)}{\sum_{y_i'=1}^c p_N(\boldsymbol{\mu}_{z_i}|\bar{\boldsymbol{\mu}}_{y_i'}, \bar{\boldsymbol{\Sigma}}_{y_i'})p(y_i')}, \quad p_{\boldsymbol{\theta},\phi}(z_i|\mathbf{v}_i) = \frac{p_N(\mathbf{v}_i|\boldsymbol{\mu}_{z_i}, \boldsymbol{\Sigma}_{z_i})\pi_{z_i}^i}{\sum_{z_i'=1}^r p_N(\mathbf{v}_i|\boldsymbol{\mu}_{z_i'}, \boldsymbol{\Sigma}_{z_i'})\pi_{z_i'}^i} \tag{4}$$

where $\boldsymbol{\theta}$ are the parameters of encoder $f_{\boldsymbol{\theta}}$, $\boldsymbol{\phi}$ includes our added distribution parameters. To reduce complexity, in this work, we investigate the feasibility of using isotropic Gaussian with tied variance, *i.e.*, $\boldsymbol{\Sigma}_{z_i} = \mathbf{I}\sigma^2$ and $\bar{\boldsymbol{\Sigma}}_{y_i} = \mathbf{I}\bar{\sigma}^2$, which turned out to be efficient in our experiments. Here, $\mathbf{I}$ is the identity matrix, $\sigma$ and $\bar{\sigma}$ are hyperparameters. Empirically, it is sufficient to set $\bar{\sigma} = 1$ and tune $\sigma$ to adjust the relativity between super- and subclasses. Also, tied variance is a commonly used trick in Gaussian discriminate analysis (GDA) for generative classifiers (Lee et al., 2018).

Finally, substituting the factors in Eq. (1) with Eq. (4), whose probabilities are specified and parameterized, we get our SCGM induced loss $\ell_{\text{SCGM}}(\mathcal{D}_{\text{train}}; \boldsymbol{\theta}, \boldsymbol{\phi})$, where $\boldsymbol{\phi} = \{\{\boldsymbol{\mu}_j\}_{j=1}^r, \{\bar{\boldsymbol{\mu}}_j\}_{j=1}^c\}$.

## 3.2 Model Optimization via Expectation-Maximization

It is hard to directly optimize $\ell_{\text{SCGM}}(\mathcal{D}_{\text{train}}; \boldsymbol{\theta}, \boldsymbol{\phi})$, because exact posterior inference is intractable (due to an exponential searching space). To solve it, we resort to variational methods, and introduce an approximated posterior $q(z_i|\mathbf{v}_i, y_i)$, where $y_i$ is included so that the inference is also conditioned on superclass. Then, a lower-bound of Eq. (1) is derived as (the details are in Appendix A.2)

$$\ell_{\text{SCGM}}(\mathcal{D}_{\text{train}}; \boldsymbol{\theta}, \boldsymbol{\phi}) \geq \frac{1}{n}\sum_{i=1}^n \mathbb{E}_{q(z_i|\mathbf{v}_i, y_i)}\big[\log p_{\boldsymbol{\phi}}(y_i|z_i) + \log p_{\boldsymbol{\theta},\boldsymbol{\phi}}(z_i|\mathbf{v}_i) - \log q(z_i|\mathbf{v}_i, y_i)\big] \tag{5}$$

which can be maximized by alternately inferring the posterior $q(z_i|\mathbf{v}_i, y_i)$ and solving the model parameters $\{\boldsymbol{\theta}, \boldsymbol{\phi}\}$ through an Expectation-Maximization (EM) algorithm.

**E-step.** This step is to infer $q(z_i|\mathbf{v}_i, y_i)$ when fixing model parameters. A straightforward way is to apply $k$-means on the embeddings $\mathbf{v}_i$, similar to (Caron et al., 2018). However, as discussed in (Asano et al., 2020), alternately applying $k$-means and learning embeddings without any constraints will lead to a degenerate solution, *i.e.*, all samples are assigned to a single (arbitrary) subclass. Also, $k$-means is unaware of the superclass $y_i$ as it doesn't optimize Eq. (5). To address this problem, inspired by (Asano et al., 2020), let $\mathbf{Q}_{z_i,i} = q(z_i|\mathbf{v}_i, y_i)\frac{1}{n}$ and $\mathbf{P}_{z_i,i} = p_{\boldsymbol{\phi}}(y_i|z_i)p_{\boldsymbol{\theta},\boldsymbol{\phi}}(z_i|\mathbf{v}_i)$ be two $r \times n$ matrices, we enforce an equal partition of subclasses by constraining $\mathbf{Q}$ to belong to the *transportation polytope* (Appendix A.3), and rewrite Eq. (5) as (the derivation is in Appendix A.4)

$$\min_{\mathbf{Q} \in \mathcal{Q}} -\Big(\text{Tr}(\mathbf{Q}^\top \log \mathbf{P}) + \frac{1}{\lambda}H(\mathbf{Q})\Big), \quad \text{where } \mathcal{Q} = \Big\{\mathbf{Q} \in \mathbb{R}_+^{r \times n}|\mathbf{Q}\mathbf{1}_n = \frac{1}{r}\mathbf{1}_r, \mathbf{Q}^\top\mathbf{1}_r = \frac{1}{n}\mathbf{1}_n\Big\} \tag{6}$$

where $H(\mathbf{Q}) = -\sum_{ij} \mathbf{Q}_{ij}\log\mathbf{Q}_{ij}$ is the entropy function, $\mathbf{1}_n$ is the vector of ones in dimension $n$, and $\lambda \geq 1$ controls the regularization (it also further relaxes the lower-bound in Eq. (5)).

The problem in Eq. (6) is a regularized instance of the optimal transport problem, whose minimizer can be written as $\mathbf{Q}^* = \text{Diag}(\mathbf{u})\mathbf{P}^\lambda\text{Diag}(\mathbf{v})$, where $\mathbf{u} \in \mathbb{R}^r$ and $\mathbf{v} \in \mathbb{R}^n$ are renormalization vectors. These vectors can be efficiently solved by the iterative Sinkhorn-Knopp algorithm (Cuturi, 2013). Once $\mathbf{Q}^*$ is found, rounding its values and using discrete codes adds more algorithmic stability.

**M-step.** Fixing $q(z_i|\mathbf{v}_i, y_i)$, the model parameters $\{\boldsymbol{\theta}, \boldsymbol{\phi}\}$ can be efficiently solved by stochastic gradient descent (SGD). Combining Eq. (4) and Eq. (5), the loss to minimize is (Appendix A.5)

$$\ell_{\boldsymbol{\phi},\boldsymbol{\theta}} = -\frac{1}{n}\sum_{i=1}^n q(z_i|\mathbf{v}_i, y_i)\Bigg[\log\frac{\exp(\boldsymbol{\mu}_{z_i}^\top \cdot \bar{\boldsymbol{\mu}}_{y_i}/\bar{\sigma}^2)p(y_i)}{\sum_{y_i'=1}^c \exp(\boldsymbol{\mu}_{z_i}^\top \cdot \bar{\boldsymbol{\mu}}_{y_i'}/\bar{\sigma}^2)p(y_i')} + \log\frac{\exp(\mathbf{v}_i^\top \cdot \boldsymbol{\mu}_{z_i}/\sigma^2)\pi_{z_i}^i}{\sum_{z_i'=1}^c \exp(\mathbf{v}_i^\top \cdot \boldsymbol{\mu}_{z_i'}/\sigma^2)\pi_{z_i'}^i}\Bigg] \tag{7}$$

It is noteworthy that the two terms in the bracket have similar form to InfoNCE (Oord et al., 2018). Therefore, Eq. (7) can be interpreted from a contrastive perspective, where the positive pairs in both terms are specified by $q(z_i|\mathbf{v}_i, y_i)$ from E-step, *i.e.*, $(z_i, y_i)$ and $(z_i, \mathbf{v}_i)$ that induce the maximal $q(z_i|\mathbf{v}_i, y_i)$, and the variances $\sigma^2$ and $\bar{\sigma}^2$ resemble temperature scaling. Thus, the M-step seeks to push the embedding $\mathbf{v}_i$ to its closest subclass $\boldsymbol{\mu}_{z_i}$, which should meanwhile be close to its superclass $\bar{\boldsymbol{\mu}}_{y_i}$, in contrast to the distances of the negative pairs. In the second term, $\pi_{z_i}^i$ (Eq. (2)) serves as a weight to select $\boldsymbol{\mu}_{z_i}$ that is close to $\bar{\boldsymbol{\mu}}_{y_i}$. As such, our problem can be seen as a generalized InfoNCE for subclass- and superclass-level contrastive learning.

In practice, we found it is beneficial to integrate Eq. (7) with cross-entropy loss $\ell_{CE}$ (on superclass)

$$\ell(\mathcal{D}_{\text{train}}; \boldsymbol{\theta}, \boldsymbol{\phi}) = \ell_{CE}(\mathcal{D}_{\text{train}}; \boldsymbol{\theta}, \{\bar{\boldsymbol{\mu}}_j\}_{j=1}^c) + \gamma\ell_{\boldsymbol{\phi},\boldsymbol{\theta}}(\mathcal{D}_{\text{train}}; \boldsymbol{\theta}, \boldsymbol{\phi}) \tag{8}$$

where $\ell_{CE}$ uses superclass means $\{\bar{\boldsymbol{\mu}}_j\}_{j=1}^c$ as the classification weights, $\gamma$ is a trade-off parameter. The detailed algorithm for training SCGM can be found in Algorithm 1 in Appendix A.6.

## 4 EXPERIMENTS

In this section, we first evaluate SCGM on benchmark datasets and compare it with SOTA methods. Then we evaluate SCGM for a model personalization task on a real-life medical dataset.

The table below summarizes the benchmark datasets: (1) BREEDS (Santurkar et al., 2020) includes four datasets {Living17, Nonliving26, Entity13, Entity30} derived from ImageNet with class hierarchy calibrated so that classes on the same level are of similar visual granularity (*i.e.*, the granularity of images for distinguishing classes); (2) CIFAR-100 (Krizhevsky, 2009); and (3) *tiered*ImageNet (Ren et al., 2018), which has 34 superclasses as per the ImageNet hierarchy, and was divided into 20/6/8 splits for (disjoint) train/val/test sets. For BREEDS and CIFAR-100, the val set is 10% of the train set. These datasets have been used in (Bukchin et al., 2021).

| Dataset | Living17 | Nonliving26 | Entity13 | Entity30 | CIFAR-100 | Tiered |
|---------|----------|-------------|----------|----------|-----------|--------|
| # Coarase classes | 17 | 26 | 13 | 30 | 20 | 20/6/8 |
| # Fine classes | 68 | 104 | 260 | 240 | 100 | 351/97/160 |
| # Train images | 88K | 132K | 334K | 307K | 50K | 448K |
| # Test images | 3.4K | 5.2K | 13K | 12K | 10K | 206K |
| Image resolution | 224×224 | 224×224 | 224×224 | 224×224 | 32×32 | 84×84 |

### 4.1 EXPERIMENTAL SETUP

**Baselines.** We compare SCGM with the most relevant SOTA models on embedding learning, including self-supervised models: (1) MoCo-v2 (Chen et al., 2020b) trained on each of the above dataset; (2) MoCo-v2-ImageNet pretrained on ImageNet by Chen et al. (2020b); (3) SwAV-ImageNet is a pretrained model by (Caron et al., 2020), which takes advantage of cluster-level contrastive learning. Note that pretrained models on full ImageNet saw more data during training than did SCGM. (4) ANCOR (Bukchin et al., 2021), as introduced in Sec.1 and 2; (5) GEORGE (Sohoni et al., 2020), a three-step framework that uses a separate clustering step to pseudo-fine-label the dataset; and (6) SeLa (Asano et al., 2020), an unsupervised algorithm for joint clustering and embedding. The detailed setup of these models was deferred to Appendix C.1.

Similar to (Bukchin et al., 2021), we include natural baselines trained on superclasses: (7) "Coarse", uses encoder $f_{\boldsymbol{\theta}}$ followed by a classifier $C$; (8) "Coarse+", uses $f_{\boldsymbol{\theta}} \to \mathcal{E} \to C$, which adds an embedder $\mathcal{E}$. Also, baselines trained on subclass labels represent performance upper-bounds: (9) "Fine", uses $f_{\boldsymbol{\theta}} \to C$; and (10) "Fine+", uses $f_{\boldsymbol{\theta}} \to \mathcal{E} \to C$. For our method, since it is a flexible framework, we tested it with a generic encoder, and a momentum-based encoder with superclass-wise dictionaries similar to ANCOR (Bukchin et al., 2021). We denote these two variants by SCGM-G and SCGM-A, where the training loss (*i.e.*, Eq. (8)) of the latter adds an InfoNCE term with angular normalization. The detailed implementation of SCGM-A can be found in Appendix C.2.

**Implementation.** The encoder is ResNet-50 (He et al., 2016) for the 224×224 images in BREEDS, and ResNet-12 for the small resolution images in CIFAR-100 and *tiered*ImageNet, as is common in few-shot learning works. The output dimension of these networks is 2048 and 640, respectively. Similar to (Chen et al., 2020b), two types of embedder $\mathcal{E}$ were tested: (1) *fc* layer: $d \to e$, and (2) MLP: $d \to d \to e$, with $e = 128$. At test time, some models performed better with $\mathcal{E}$ dropped while some were better otherwise. For each model, the better case was reported. For training, we used cosine annealing with warm restarts schedule (Loshchilov & Hutter, 2017) with 20 epochs per cycle. The batch size was 256 for BREEDS, 1024 for CIFAR-100, and 512 for *tiered*ImageNet. The learning rate was 0.03 for BREEDs, and 0.12 for CIFAR-100 and *tiered*ImageNet. The weight decay was $1e^{-4}$. All models were trained with 200 epochs. The best training practices of (Tian et al., 2020) were also applied. For SCGM, we set $\gamma = 0.5$, $\sigma^2 = 0.1$, and $\lambda = 25$ ($\lambda$ follows (Asano et al., 2020)). The number of latent variables $r$ varies w.r.t. different datasets, and was grid-searched from 100 to 500 with step size 50. All hyperparameters were set according to validation sets.

### 4.2 EXPERIMENTAL RESULTS

Following (Tian et al., 2020), we report the mean accuracy and the 95% confidence interval of 1000 random episodes with 5-way/all-way $k$-shot, 15-query tests. Unless otherwise stated, $k = 1$. Effects of more shots are in Appendix D.1. For each episode, 5 augmented copies were created per support sample. A logistic regression classifier was trained on the support set embedded by each model with

| Method | Living17 | | Nonliving26 | | Entity13 | | Entity30 | |
|---|---|---|---|---|---|---|---|---|
| | 5-way | All-way | 5-way | All-way | 5-way | All-way | 5-way | All-way |
| Fine (upper-bounds) | 91.10±0.47 | 58.95±0.16 | 85.25±0.49 | 47.68±0.13 | 91.01±0.39 | 50.19±0.08 | 91.65±0.41 | 56.54±0.09 |
| Fine+ (upper-bounds) | 90.75±0.48 | 62.65±0.18 | 90.33±0.47 | 60.68±0.14 | 94.72±0.33 | 65.18±0.09 | 94.02±0.36 | 63.72±0.10 |
| MoCo-v2 | 56.66±0.70 | 18.57±0.11 | 63.51±0.75 | 21.07±0.11 | 82.00±0.67 | 33.06±0.07 | 80.37±0.62 | 28.62±0.06 |
| MoCo-v2-ImageNet | 83.85±0.65 | 40.59±0.15 | 77.02±0.72 | 35.13±0.11 | 85.07±0.61 | 35.78±0.07 | 83.44±0.64 | 32.00±0.08 |
| SwAV-ImageNet | 81.15±0.63 | 39.57±0.15 | 75.67±0.70 | 35.28±0.12 | 81.76±0.61 | 34.53±0.07 | 80.76±0.63 | 31.92±0.08 |
| GEORGE | 85.50±0.63 | 32.10±0.11 | 80.75±0.62 | 31.07±0.10 | 77.68±0.68 | 15.58±0.05 | 83.41±0.59 | 21.12±0.06 |
| SeLa | 52.46±0.68 | 14.87±0.10 | 54.08±0.72 | 14.23±0.07 | 63.29±0.73 | 13.83±0.05 | 60.68±0.71 | 12.20±0.05 |
| Coarse | 85.12±0.74 | 33.83±0.10 | 83.53±0.64 | 33.52±0.11 | 82.33±0.61 | 17.49±0.04 | 87.03±0.54 | 24.01±0.06 |
| Coarse+ | 79.29±0.65 | 37.44±0.12 | 75.91±0.66 | 36.80±0.11 | 83.23±0.66 | 31.15±0.07 | 84.81±0.61 | 33.22±0.08 |
| ANCOR | 89.23±0.55 | 45.14±0.12 | 86.23±0.54 | 43.10±0.11 | **90.58±0.54** | **42.29±0.08** | 88.12±0.54 | 41.79±0.08 |
| ANCOR-*fc* | 90.41±0.57 | 46.19±0.16 | 88.77±0.54 | 45.34±0.13 | 89.05±0.58 | 38.52±0.08 | 91.84±0.49 | 42.33±0.10 |
| SCGM-G | 89.72±0.54 | 48.74±0.15 | **89.87±0.51** | **49.25±0.13** | 90.15±0.51 | 40.00±0.08 | **92.90±0.46** | 42.17±0.08 |
| SCGM-A | **90.97±0.55** | **49.31±0.16** | 88.78±0.55 | 46.93±0.13 | 88.48±0.59 | 41.07±0.09 | 91.22±0.51 | **44.14±0.09** |

Table 1: Comparison on BREEDS datasets. Bold and underlined numbers are the best and second best results.

| Method | 5-way | All-way |
|---|---|---|
| Fine | 74.36±0.68 | 28.82±0.11 |
| Fine+ | 75.53±0.68 | 31.35±0.11 |
| MoCo-v2 | 48.07±0.68 | 10.61±0.06 |
| GEORGE | 70.64±0.70 | 24.54±0.10 |
| SeLa | 38.52±0.60 | 5.87±0.05 |
| Coarse | 74.40±0.70 | 27.37±0.11 |
| Coarse+ | 70.69±0.69 | 26.16±0.10 |
| ANCOR | 74.56±0.70 | 29.84±0.11 |
| ANCOR-*fc* | 74.73±0.73 | 27.32±0.10 |
| SCGM-G | 76.19±0.73 | **29.92±0.11** |
| SCGM-A | **77.37±0.77** | 25.91±0.10 |

Table 2: Comparison on CIFAR-100 datasets. Bold and underlined numbers are the best and second best results.

| Method | Living17 | Nonliving26 | Entity13 | Entity30 |
|---|---|---|---|---|
| Fine (upper-bounds) | 66.60±0.89 | 66.33±0.92 | 64.20±0.63 | 65.74±0.68 |
| Fine+ (upper-bounds) | 70.72±0.92 | 74.02±0.91 | 72.24±0.60 | 73.86±0.67 |
| MoCo-v2 | 41.38±0.62 | 45.21±0.78 | 37.78±0.49 | 41.67±0.59 |
| MoCo-v2-ImageNet | 51.73±0.79 | 53.69±0.94 | 42.78±0.54 | 45.13±0.60 |
| SwAV-ImageNet | 52.26±0.73 | 54.77±0.90 | 42.95±0.54 | 45.78±0.60 |
| GEORGE | 38.17±0.52 | 43.55±0.69 | 21.42±0.30 | 30.79±0.47 |
| SeLa | 40.58±0.61 | 42.56±0.70 | 23.90±0.32 | 31.02±0.46 |
| Coarse | 38.14±0.53 | 40.38±0.62 | 16.01±0.22 | 28.20±0.44 |
| Coarse+ | 40.70±0.59 | 50.21±0.83 | 37.13±0.48 | 43.10±0.65 |
| ANCOR | 48.77±0.71 | 49.64±0.88 | 42.00±0.47 | 45.17±0.59 |
| ANCOR-*fc* | 51.07±0.82 | 53.51±1.00 | 41.83±0.48 | 47.82±0.65 |
| SCGM-G | 53.48±0.81 | **57.32±1.04** | 43.89±0.58 | 46.80±0.78 |
| SCGM-A | **53.88±0.90** | 55.12±1.00 | **45.09±0.58** | **50.02±0.71** |

Table 3: Intra-class few-shot learning results of the compared methods on BREEDS datasets.

$L_2$ norm. The model with the resulting classifier was used to classify the query samples. The test classes of each episode were a random subset of all subclasses (or all of them for all-way tests). Two cases were evaluated: (1) unseen subclasses of *seen* superclasses (on BREEDS, CIFAR-100); and (2) unseen subclasses of *unseen* superclasses (on *tiered*ImageNet). On CIFAR-100 and *tiered*ImageNet, ImageNet-pratrained models were excluded since pretrained ResNet-12 models are unavailable.

**Evaluation case (1).** We first evaluate when test classes are unseen subclasses of the training superclasses. Table 1 and 2 summarize the results. For ANCOR, its variants when the embedder $\mathcal{E}$ is *fc* (with '-*fc*') or MLP (without '-*fc*') were included as either variant does not consistently outperform another. The embedder setup of all models are in Appendix C.1. From the tables, we have several observations. First, models that cannot leverage superclasses, such as MoCo-v2 and SeLa, are inferior to Coarse(+). Although full ImageNet-pretrained models used larger training datasets, they are unable to fill the gap of superclass supervision, compared with Coarse(+). Second, the limitation of GEORGE indicates a separate clustering step may induce misleading pseudo-labels that degenerate embedding quality. Third, ANCOR and SCGM consistently outperform Coarse(+), and SCGM significantly outperforms ANCOR(-*fc*) in most all-way cases. With a generic encoder, SCGM-G is suprior or competitive to ANCOR(-*fc*). Using momentum contrast, SCGM-A further gains on some datasets. Note that 5-way has less room to improve than the all-way cases, because the random subclasses are more likely to be from different superclasses, which may degrade the evaluation to coarse classification. The results suggest SCGM could perform better when more subclasses are from the same superclass. To investigate it, we evaluated an "intra-class" case when all subclasses of a random superclass were sampled in each episode. Table 3 indicates SCGM got more improvements in this challenging task, implying its embeddings distinguish subclasses better than other models.

Moreover, Table 4 compares the computational costs of ANCOR and SCGM-G, from which we can see training ANCOR requires much more resources and time than SCGM-G as it inherits MoCo-v2 (Chen et al., 2020b). Also, its memory costs vary a lot

| Dataset | ANCOR | | | SCGM-G | | |
|---|---|---|---|---|---|---|
| | memory | time (h) | size | memory | time (h) | size |
| Living17 | 51.98G | 16.43 | 0.74G | 29.63G | 8.09 | 0.09G |
| Nonliving26 | 55.27G | 25.42 | 1.02G | 29.63G | 12.94 | 0.09G |
| Entity13 | 50.35G | 65.45 | 0.62G | 29.63G | 43.94 | 0.09G |
| Entity30 | 56.91G | 58.98 | 1.15G | 29.63G | 34.56 | 0.09G |

Table 4: Computational costs on 4 Quadro RTX6000 24G GPUs.

| Method | 5-way-1-shot | 5-way-5-shot | All-way-1-shot |
|---|---|---|---|
| Fine | 70.15±0.70 | 84.96±0.47 | 15.42±0.06 |
| Fine+ | 67.33±0.69 | 84.43±0.47 | 15.22±0.06 |
| MoCo-v2 | 53.19±0.68 | 70.90±0.58 | 8.33±0.04 |
| GEORGE | 60.97±0.70 | 77.72±0.55 | 10.24±0.05 |
| SeLa | 47.96±0.63 | 66.39±0.57 | 7.17±0.04 |
| C2F-VSME | 60.54±0.79 | 75.22±0.63 | N/A |
| Coarse | 61.61±0.74 | 75.72±0.56 | 8.42±0.04 |
| Coarse+ | 64.07±0.75 | 78.41±0.58 | 10.70±0.05 |
| ANCOR | 62.86±0.70 | 79.93±0.52 | 11.97±0.06 |
| ANCOR-*fc* | 60.43±0.69 | 78.18±0.53 | 11.04±0.05 |
| SCGM-G | **64.64±0.71** | **80.83±0.52** | **12.32±0.05** |
| SCGM-A | 64.47±0.71 | 80.68±0.51 | 11.75±0.05 |

Table 5: Comparison on *tiered*ImageNet.

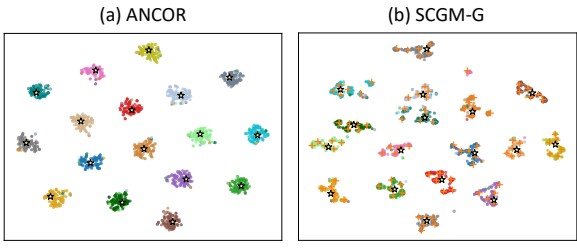

(a) ANCOR        (b) SCGM-G

Figure 2: The tSNE visualization of the embeddings learned by ANCOR and SCGM-G. Star represents superclass means. "+" marker represents subclass means of SCGM-G.

by datasets since its dictionaries are superclass-wise whose size changes w.r.t. the number of super-classes. In contrast, SCGM-G had lower and stable costs, with better performance, thus is more favorable in practical deployments.

**Evaluation case (2).** Since *tiered*ImageNet has distinct train/val/test sets, it was used to evaluate the case when the testing subclasses is out of any training superclass. All models were trained with the superclasses of the train set, and tested on the subclass labels of the test set. Table 5 presents the results. Another three-step method C2F-VSME (Yang et al., 2021) was included as the same evaluation has been performed (but their code is unavailable). First, from the smaller gaps between Fine(+) and Coarse(+), this task is more difficult than the previous case. Similar to Table 1, MoCo-v2 and SeLa are inferior to Coarse(+), underlining the benefit of superclass supervision, even when test subclasses are not their descendants. Compared to GEORGE and C2F-VSME, ANCOR and SCGM can better leverage superclasses, by using coherent learning frameworks. Both SCGM-G and SCGM-A outperform ANCOR(-*fc*) in most cases, demonstrating the effectiveness of explicit modeling of latent subclasses, in both evaluation case (1) and (2).

**Visualization of embedding.** To understand how SCGM uncovers the subclass structures, we visualize a random batch of embeddings using tSNE (Maaten & Hinton, 2008). Fig. 2 presents the visualization results of ANCOR and SCGM-G on Living17 dataset, where colors mark different superclasses, "+" markers are the learned subclass means by SCGM-G, and stars represent superclass means. ANCOR's embeddings resemble those of Coarse+ (Appendix D.5) as it distinguishes subclasses from an angular perspective, which induces intra-class variation that is suboptimal. SCGM-G explicitly detects the boundaries of both super- and subclasses, properly positions Gaussian means, and associates subclasses with their corresponding superclasses, whereby establishing a hierarchical structure. We evaluated SCGM-G's detected subclasses using the subclass labels in the val set. It got an NMI of 71.58% and purity accuracy of 74.79%, much higher than its unsupervised counterpart SeLa (NMI 36.50%, purity 38.14%). This validates SCGM's effectiveness in leveraging superclass guidance for subclass inference. Several other model's visualization can be found in Appendix D.5.

**Performance analysis.** We have evaluated SCGM in terms of the effects of more shots, the impacts of the number of latent variables $r$, the impacts of the variance $\sigma^2$, and the convergence, which are in Appendix D.1, D.2, D.3, D.4.

### 4.3 CASE STUDY: PERSONALIZED PREDICTION ON MEDICAL RECORDS

Next, we evaluate SCGM on a real Dialysis-Event dataset collected by several hospitals in Japan. This dataset consists of one-year medical records of 673 hemodialysis patients. It contains 40 temporal features (*e.g.*, blood flow, venous pressure, *etc.*) that were monitored during each hemodialysis session (thrice/week). Each session was annotated to indicate whether certain subtypes of events have occurred. A subtype corresponds to certain unstable behaviors of blood pressure during the dialysis, as categorized by experts. As introduced in Sec. 1, these events are risky and may develop complications, thus should be avoided. Formally, our task is as follows. Given $\mathbf{x}^i_{t:t+w-1} \in \mathbb{R}^{d \times w}$, a length-$w$ segment including the $i$-th patient's records starting at time step $t$, where $d = 40$ is the dimensionality, the task outcome is a label $y \in \{0, 1\}$ indicating event occurs ($y = 1$) or not ($y = 0$), at time step $t + w$. When $y = 1$, a fine-grained label $\hat{y} \in \{1, ..., s\}$ is to be predicted to mark the subtype of the event.

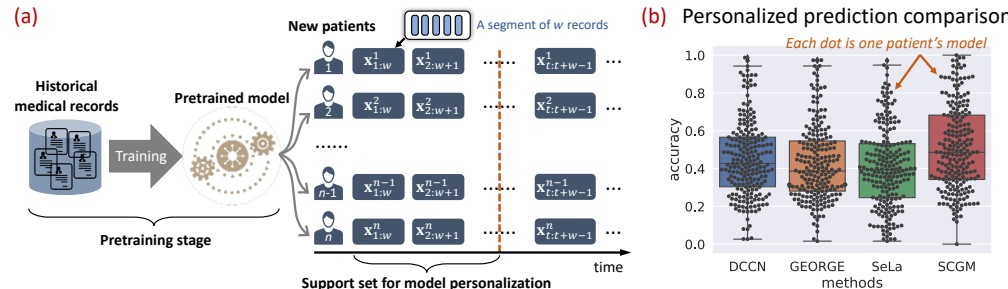

Figure 3: (a) An illustration of the model personalization framework for fine-grained event prediction for hemodialysis patients. (b) The distributions of patient-wise prediction accuracy on the query set by different methods. Each dot represents the accuracy of one patient's personalized model.

Since annotating fine-grained labels is costly, training a model on them at a large scale is infeasible in practice. To predict event subtypes, we use a *model personalization* framework as illustrated in Fig. 3(a). At the pre-training stage, only binary labels are used. For every new patient, a short period of (a few) medical records are collected as the support set for the fine-grained labels w.r.t. that specific patient. It is used to train a personalized classifier based on the pre-trained model.

To evaluate how SCGM fits this paradigm, we randomly split the 673 patients into 50%/50% train/test set, and reserved 10% of the train set for validation. For each testing patient, the first 50% records (*i.e.*, half year) form the support set, and the remaining form the query set. It is likely some event types didn't appear in the support set will appear in the query set, which are out-of-distribution (OOD) cases, thus is out of our scope. Hence, we only selected patients without such cases, which constitute a major part (>65%) of the test set. The dataset has two event subtypes, *i.e.* $s = 2$. The segments $\mathbf{x}_{t:t+w-1}^{(i)}$ were sampled with $w = 5$, stride 1. The encoder is a Dual-Channel Combiner Network (DCCN), similar to (Che et al., 2016). DCCN has two channels (*e.g.*, RNNs and MLP) for encoding temporal and static features, respectively. The outputs of them are integrated by a classification head. Its architecture can be found in Appendix C.3. In this application, each patient has 28 static features such as demographic information and infrequent blood test results.

Table 6 presents the average accuracy of all testing patients over 10 random train/test splits for the applicable methods. As is consistent with Sec. 4.2, unsupervised method SeLa and three-step framework GEORGE cannot improve DCCN's performance due to (1) ignorance of superclass supervision, and (2) a separate pseudo-labeling step, respectively. In contrast, SCGM outperforms the original DCCN significantly. Fig. 3(b) compares different models by their distributions of personalized prediction accuracy of different testing patients in a random train/test split. Each dot represents the mean accuracy of one patient's personalized predictions in the

| Method | Accuracy (%) |
|--------|--------------|
| DCCN | 45.59±0.99 |
| GEORGE | 43.63±0.63 |
| SeLa | 42.32±0.80 |
| SCGM | **49.89±0.90** |

Table 6: The comparison results on Dialysis-Event dataset.

query set (220 dots in total). From the figure, SCGM has more dots close to the top (*e.g.*, above 0.8) than other models, meaning more patients received fairly accurate predictions on the fine-grained labels. Although current performance could be limited by the data size, the results demonstrate the flexibility of SCGM, and its potential in tackling the CGFS problem in real practice.

## 5 CONCLUSION

In this paper, we proposed a new method, superclass-conditional Gaussian mixture model (SCGM), that underlies a general framework for tackling CGFS across domains. SCGM imitates the generative process of samples from hierarchies of classes, and explicitly models unobserved subclasses by latent variables. It dynamically computes a Gaussian mixture for every sample given its superclass, with a small overhead to an encoder. We learned the model parameters by maximum likelihood estimation in a principled Expectation-Maximization framework, and also theoretically linked SCGM with contrastive learning. The extensive experiments on various datasets demonstrated the effectiveness, efficiency, and flexibility of our proposed method.

## 6  ETHICS STATEMENT

In the case study in Sec. 4.3, we evaluated our SCGM on datasets related to human subjects that are hemodialysis patients. We clarify that we got the permission to perform the experiments on the dataset as described in Sec. 4.3. There is no ethics issue nor harmful insights in our current experiments. The dataset was properly used under the guidance of the data providers and domain experts. Sensitive information about the patients was well protected. The final description about the experiments was checked and approved by the data providers and relevant authorities.

## 7  REPRODUCIBILITY STATEMENT

The code of SCGM is available at https://github.com/nijingchao/SCGM for reproducibility study. The benchmark datasets are large and cannot be upooaded. The repository includes an instruction on how to obtain the datasets and process them. Relevant instructions were provided on how to use the code to train SCGM-G and SCGM-A, and evaluate their performance for CGFS task in 5-way and all-way cases (Table 1, 2, 5), and in the intra-class case (Table 3).

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

# A APPENDIX FOR DETAILS OF DERIVING SCGM

## A.1 DERIVATION OF PARAMETERIZED PROBABILITIES

In this section, we provide the details of the probabilities in Eq. (4). First, using Bayesian rules and marginal property, we can derive

$$p(y_i|z_i) = \frac{p(z_i|y_i)p(y_i)}{p(z_i)} = \frac{p(z_i|y_i)p(y_i)}{\sum_{y_i'=1}^{c} p(z_i|y_i')p(y_i')}, \quad p(z_i|\mathbf{v}_i) = \frac{p(\mathbf{v}_i|z_i)\pi_{z_i}^i}{p(\mathbf{v}_i)} = \frac{p(\mathbf{v}_i|z_i)\pi_{z_i}^i}{\sum_{z_i'=1}^{r} p(\mathbf{v}_i|z_i')\pi_{z_i'}^i} \quad (9)$$

Since $\boldsymbol{\mu}_{z_i}$ and $\bar{\boldsymbol{\mu}}_{y_i}$ are surrogate representations of $z_i$ and $y_i$ in the embedding space, substituting $p(z_i|y_i)$ and $p(\mathbf{v}_i|z_i)$ with the equality in Eq. (3), Eq. (4) can be achieved.

## A.2 THE LOWER-BOUND OF THE LIKELIHOOD FUNCTION

In this section, we provide the details of the lower-bound in Eq. (5). By introducing the approximated posterior $q(z_i|\mathbf{v_i}, y_i)$, the likelihood in Eq. (1) becomes

$$
\begin{aligned}
\ell(\mathcal{D}_{\text{train}}; \boldsymbol{\theta}) &= \frac{1}{n} \sum_{i=1}^{n} \log \left[ \sum_{z_i=1}^{r} p(y_i|z_i)p_{\boldsymbol{\theta}}(z_i|\mathbf{v_i}) \frac{q(z_i|\mathbf{v_i}, y_i)}{q(z_i|\mathbf{v_i}, y_i)} \right] \\
&= \frac{1}{n} \sum_{i=1}^{n} \log \left[ \sum_{z_i=1}^{r} q(z_i|\mathbf{v_i}, y_i) \frac{p(y_i|z_i)p_{\boldsymbol{\theta}}(z_i|\mathbf{v_i})}{q(z_i|\mathbf{v_i}, y_i)} \right] \\
&\geq \frac{1}{n} \sum_{i=1}^{n} \sum_{z_i=1}^{r} q(z_i|\mathbf{v_i}, y_i) \Big[ \log p(y_i|z_i) + \log p_{\boldsymbol{\theta}}(z_i|\mathbf{v_i}) - \log q(z_i|\mathbf{v_i}, y_i) \Big] \\
&= \frac{1}{n} \sum_{i=1}^{n} \mathbb{E}_{q(z_i|\mathbf{v}_i, y_i)} \Big[ \log p_{\boldsymbol{\phi}}(y_i|z_i) + \log p_{\boldsymbol{\theta}, \boldsymbol{\phi}}(z_i|\mathbf{v_i}) - \log q(z_i|\mathbf{v}_i, y_i) \Big]
\end{aligned}
\quad (10)
$$

where the third step uses Jensen's inequality. This completes the derivation of Eq. (5).

## A.3 DETAILS ON TRANSPORTATION POLYTOPE

Let $\mathbf{a}$ be a probability vector in the simplex $\Delta_r = \{\mathbf{x} \in \mathbb{R}_+^r | \mathbf{x}^\top \mathbf{1}_r = 1\}$ and $\mathbf{b}$ be a probability vector in the simplex $\Delta_n = \{\mathbf{x} \in \mathbb{R}_+^n | \mathbf{x}^\top \mathbf{1}_n = 1\}$, where $\mathbf{1}_r$ and $\mathbf{1}_n$ are the vectors of ones in dimension $r$ and $n$, respectively. The transportation polytope of $\mathbf{a}$ and $\mathbf{b}$, *i.e.*, the polyhedral set of $r \times n$ matrices, is defined as (Cuturi, 2013)

$$\mathcal{Q} = \left\{ \mathbf{Q} \in \mathbb{R}_+^{r \times n} | \mathbf{Q}\mathbf{1}_n = \mathbf{a}, \mathbf{Q}^\top \mathbf{1}_r = \mathbf{b} \right\} \quad (11)$$

which means $\mathcal{Q}$ is a set of non-negative $r \times n$ matrices with row and column sums $\mathbf{a}$ and $\mathbf{b}$, respectively. From a probabilistic perspective, any matrix $\mathbf{Q} \in \mathcal{Q}$ represents a joint probability of two multinomial random variables $X$ and $Y$ that take values in $\{1, ..., r\}$ and $\{1, ..., n\}$ with marginal distributions $\mathbf{a}$ and $\mathbf{b}$, respectively.

The transportation polytope defines a space of transport matrix $\mathbf{Q}$ for mapping $\mathbf{a}$ to $\mathbf{b}$. Given a cost matrix $\mathbf{M} \in \mathbb{R}_+^{r \times n}$, the cost of the mapping can be quantified as $\text{Tr}(\mathbf{Q}^\top \mathbf{M})$, and the problem

$$\min_{\mathbf{Q} \in \mathcal{Q}} \text{Tr}(\mathbf{Q}^\top \mathbf{M}) \quad (12)$$

is called an optimal transport (OT) problem between $\mathbf{a}$ and $\mathbf{b}$, given cost $\mathbf{M}$. Since the traditional algorithms to solve the OT problem scale inefficiently to large datasets, (Cuturi, 2013) introduced a fast Sinkhorn-Knopp algorithm, which amounts to solve a regularized problem

$$\min_{\mathbf{Q} \in \mathcal{Q}} \text{Tr}(\mathbf{Q}^\top \mathbf{M}) - \frac{1}{\lambda} H(\mathbf{Q}) \quad (13)$$

where $H(\mathbf{Q}) = -\sum_{ij} \mathbf{Q}_{ij} \log \mathbf{Q}_{ij}$ is the entropy function, and $\lambda$ is a trade-off parameter. If $\lambda$ is large, optimizing Eq. (13) approximates optimizing Eq. (12).

In our case of Eq. (6), $\mathbf{M} = -\log \mathbf{P}$, $\mathbf{a} = \frac{1}{r}\mathbf{1}_r$, and $\mathbf{b} = \frac{1}{n}\mathbf{1}_n$. In addition, similar to (Asano et al., 2020), $\mathbf{Q}_{z_i,i} = q(z_i|\mathbf{v}_i, y_i)\frac{1}{n}$ can be considered as a joint probability of $z_i$ and the $i$-th sample (note the condition $(\mathbf{v}_i, y_i)$ corresponds to the incidence of the $i$-th sample) with a uniform prior $\frac{1}{n}$. For the details of the Sinkhorn-Knopp algorithm, we refer the readers to (Cuturi, 2013).

It is noteworthy that our optimization problem in Eq. (6) in the E-step is derived from our likelihood function in Eq. (5), which turns out to be a form of the regularized OT problem in Eq. (13), rather than directly formulated using Eq. (13). The derivation of the optimization problem in Eq. (6) can be found in next section (Appendix A.4).

## A.4  THE OPTIMIZATION PROBLEM IN E-STEP

In this section, we elaborate the optimization problem for inferring the posterior $q(z_i|\mathbf{v_i}, y_i)$ in Eq. (6). Substituting $q(z_i|\mathbf{v_i}, y_i)$ with $n\mathbf{Q}_{z_i,i}$, and substituting $p_{\boldsymbol{\phi}}(y_i|z_i)p_{\boldsymbol{\theta},\boldsymbol{\phi}}(z_i|\mathbf{v}_i)$ with $\mathbf{P}_{z_i,i}$, the third step in Eq. (10) can be rewritten by

$$\sum_{i=1}^{n}\sum_{z_i=1}^{r}\Big[\mathbf{Q}_{z_i,i}\log\mathbf{P}_{z_i,i} - \mathbf{Q}_{z_i,i}\log\mathbf{Q}_{z_i,i} - \mathbf{Q}_{z_i,i}\log n\Big] \geq \mathrm{Tr}(\mathbf{Q}^{\top}\log\mathbf{P}) + \frac{1}{\lambda}H(\mathbf{Q}) - \log n \qquad (14)$$

where $\mathrm{Tr}(\cdot)$ is the trace function, $H(\cdot)$ is the entropy function, the last constant term is because $\sum_{i=1}^{n}\sum_{z_i=1}^{r}\mathbf{Q}_{z_i,i} = 1$, and the inequality stems from $\lambda \geq 1$, which is introduced to control the strength of the regularization (Cuturi, 2013). This is the objective function in Eq. (6), where the constant $\log n$ is omitted because it has no effects on the optimization.

The right hand side of Eq. (14) is a valid lower bound of the likelihood function in Eq. (5) according to Eq. (10). Therefore, optimizing it (*i.e.*, the objective function in Eq. (6)) provides a reasonably approximated solution to the likelihood function in Eq. (5). In addition, the transport polytope $\mathcal{Q}$ in Eq. (6) is introduced as an equal partition constraint on subclasses (Asano et al., 2020), so that the optimization problem is feasible. It is noteworthy that although subclasses have the equal partition constraint, superclasses can have different sizes since they can have different numbers of subclasses.

## A.5  THE OPTIMIZATION PROBLEM IN M-STEP

In this section, we derive the optimization problem for learning model parameters in Eq. (7). Introducing the probabilities specification in Eq. (4) to the negative of the lower-bounded likelihood in Eq. (5), and removing terms that are irrelevant to model parameters $\{\boldsymbol{\theta}, \boldsymbol{\phi}\}$, we have

$$\ell_{\boldsymbol{\theta},\boldsymbol{\phi}} = \frac{1}{n}\sum_{i=1}^{n}\sum_{z_i=1}^{r}q(z_i|\mathbf{v_i}, y_i)\Big[\log\frac{p_N(\boldsymbol{\mu}_{z_i}|\bar{\boldsymbol{\mu}}_{y_i}, \bar{\boldsymbol{\Sigma}}_{y_i})p(y_i)}{\sum_{y_i'=1}^{c}p_N(\boldsymbol{\mu}_{z_i}|\bar{\boldsymbol{\mu}}_{y_i'}, \bar{\boldsymbol{\Sigma}}_{y_i'})p(y_i')} + \log\frac{p_N(\mathbf{v}_i|\boldsymbol{\mu}_{z_i}, \boldsymbol{\Sigma}_{z_i})\pi^i_{z_i}}{\sum_{z_i'=1}^{r}p_N(\mathbf{v}_i|\boldsymbol{\mu}_{z_i'}, \boldsymbol{\Sigma}_{z_i'})\pi^i_{z_i'}}\Big] \tag{15}$$

Considering isotropic Gaussian with $\bar{\boldsymbol{\Sigma}}_{y_i} = \mathbf{I}\bar{\sigma}^2$, we have

$$p_N(\boldsymbol{\mu}_{z_i}|\bar{\boldsymbol{\mu}}_{y_i}, \bar{\boldsymbol{\Sigma}}_{y_i}) = \exp\Big(\frac{-(\boldsymbol{\mu}_{z_i} - \bar{\boldsymbol{\mu}}_{y_i})^2}{2\bar{\sigma}^2}\Big) = \exp\Big(\frac{(\boldsymbol{\mu}_{z_i}^{\top}\cdot\bar{\boldsymbol{\mu}}_{y_i} - 1)}{\bar{\sigma}^2}\Big) = \frac{\exp\big(\boldsymbol{\mu}_{z_i}^{\top}\cdot\bar{\boldsymbol{\mu}}_{y_i}/\bar{\sigma}^2\big)}{\exp\big(1/\bar{\sigma}^2\big)} \tag{16}$$

where the second equivalence is because we apply $L_2$ norm to $\boldsymbol{\mu}_{z_i}$ and $\bar{\boldsymbol{\mu}}_{y_i}$, respectively. Similarly, considering $\boldsymbol{\Sigma}_{z_i} = \mathbf{I}\sigma^2$, and applying $L_2$ norm to $\mathbf{v}_i$, we have

$$p_N(\mathbf{v}_i|\boldsymbol{\mu}_{z_i}, \boldsymbol{\Sigma}_{z_i}) = \exp\Big(\frac{-(\mathbf{v}_i - \boldsymbol{\mu}_{z_i})^2}{2\sigma^2}\Big) = \exp\Big(\frac{(\mathbf{v}_i^{\top}\cdot\boldsymbol{\mu}_{z_i} - 1)}{\sigma^2}\Big) = \frac{\exp\big(\mathbf{v}_i^{\top}\cdot\boldsymbol{\mu}_{z_i}/\sigma^2\big)}{\exp\big(1/\sigma^2\big)} \tag{17}$$

Substituting the corresponding terms in Eq. (15) with Eq. (16) and Eq. (17), and removing the constants $\exp(1/\bar{\sigma}^2)$ and $\exp(1/\sigma^2)$ in the fractions, we obtain the loss function in Eq. (7). In Eq. (7), the prior $p(y_i)$ can be set as uniform, *i.e.*, $p(y_i) = \frac{1}{c}$. If using a prior from Dirichlet distribution, $p(y_i)$ can be instantiated by estimating the parameters $\boldsymbol{\alpha}$ (of the Dirichlet distribution) by the ratio of different superclass labels in $\mathcal{D}_{\mathrm{train}}$, the training dataset. Also, $\pi^i_{z_i}$ in the second term of Eq. (7) can be calculated using Eq. (2).

## A.6  THE TRAINING ALGORITHM OF SCGM

The training algorithm of SCGM is summarized in Algorithm 1.

---

**Algorithm 1:** Superclass-conditional Gaussian mixture model (SCGM)

---

**Input:** encoder $f_{\boldsymbol{\theta}}$, training dataset $\mathcal{D}_{\text{train}}$ (number of classes is $c$), hyperparameters $r, \lambda, \gamma, \sigma$
**Output:** model parameters $\{\boldsymbol{\theta}, \boldsymbol{\phi}\}$

1  $\boldsymbol{\phi} = \text{Initialization}(c, r)$       `// initialize Gaussian means` $\boldsymbol{\phi} = \{\{\boldsymbol{\mu}_j\}_{j=1}^{r}, \{\bar{\boldsymbol{\mu}}_j\}_{j=1}^{c}\}$
2  **for** $i \leftarrow 1$ *to MaxEpoch* **do**
    `/* E-step */`
3      **for** $(\mathbf{x}, y)$ *in Dataloader($\mathcal{D}_{train}$)* **do**
       `/* load a minibatch` $(\mathbf{x}, y)$ `*/`
4         $\mathbf{v} = f_{\boldsymbol{\theta}}(\mathbf{x})$                             `// get embeddings`
5         Compute $\mathbf{p}$ for Eq. (6) based on Eq. (4) using $\mathbf{v}$ and $\boldsymbol{\phi}$
6      **end**
7      Concatenate all batch-wise probabilities $\mathbf{p}$ to get $\mathbf{P}$ for $\mathcal{D}_{\text{train}}$
8      $\mathbf{Q} = \text{SinkhornKnopp}(\mathbf{P}, \lambda)$      `// Inferring posterior using Sinkhorn-Knopp algorithm for optimizing Eq.(6)`
    `/* M-step */`
9      **for** $(\mathbf{x}, y, \mathbf{q})$ *in Dataloader($\mathcal{D}_{train}$)* **do**
       `/* load a minibatch` $(\mathbf{x}, y, \mathbf{q})$ `where` $\mathbf{q}$ `is obtained from E-step */`
10        $\mathbf{v} = f_{\boldsymbol{\theta}}(\mathbf{x})$                      `// forward pass through the encoder`
11        Calculate $\ell(\mathbf{v}, y, \mathbf{q}, \boldsymbol{\phi}, \gamma, \sigma)$   `// calculate loss in Eq.(8) using` $\ell_{\boldsymbol{\phi}, \boldsymbol{\theta}}$ `in Eq.(7)`
12        $\boldsymbol{\theta}, \boldsymbol{\phi} = \text{SGD}(\ell, \boldsymbol{\theta}, \boldsymbol{\phi})$                   `// update model parameters`
13     **end**
14 **end**

---

# B   Appendix for Further Discussion

## B.1   Discussion on the equal partition constraint

The equal partition constraint on subclasses in Eq. (6) theoretically assumes subclasses are of equal or similar sizes, through using $\mathbf{Q}\mathbf{1}_n = \frac{1}{r}\mathbf{1}_r$ (i.e., $\mathbf{a} = \frac{1}{r}\mathbf{1}_r$ in Eq. (11)) as a prior distribution of the subclass sizes (superclasses can have different sizes because they can have different numbers of subclasses). It is possible to replace $\frac{1}{r}\mathbf{1}_r$ by other prior distributions to reflect an uneven distribution of the subclasses sizes. In this work, we used a uniform prior because we don't assume such prior knowledge (since the subclasses are unobserved during training). This prior is inspired by (Asano et al., 2020). If we don't have such a prior, the searching space of subclasses will be exponential and the optimization is intractable. To make the optimization feasible, we included this constraint to help reduce the searching space. In this way, we found the solution is efficient and effective to our CGFS problem, which does not necessitate an exact identification of every subclass. Compared to other baseline methods, the performance improvements indicate that reasonable boundaries between subclasses can be learned by SCGM with this constraint for model adaptation, though the boundaries may not be ideal. However, it is worth to note that this uniform prior may not always be reasonable in different applications. For example, if identifying rare subclass is important in an application (Sohoni et al., 2020), a more reasonable prior may be considered. In our experiments, the evaluations on the benchmark datasets (which were used for a consistent setup with the existing works (Bukchin et al., 2021)) do not reflect the case of substantially skew distributions of subclass sizes. We leave the extension of the proposed method to this scenario in our future work.

## B.2   Discussion on the evaluation cases in Section 4.2

In Section 4.2, we evaluated two fundamental cases of (1) unseen subclasses of *seen* superclasses; and (2) unseen subclasses of *unseen* superclasses. Another case that considers the mixture of the subclasses of seen and unseen superclasses is worth discussion because it is a practical case. As discussed in (Xian et al., 2018; Oreshkin et al., 2020), a mixture of samples from seen and unseen classes during testing may result in an imbalance of prediction performance between seen and unseen classes, and the inductive bias may lead to misclassification of unseen classes towards the seen classes. It is noteworthy that (Xian et al., 2018; Oreshkin et al., 2020) focus on a generalized zero-shot learning problem, which is not for cross-granularity adaptation and don't consider class hierarchy. Thus their task is remarkably different from our CGFS task. Despite the difference, we are inspired to be aware of the challenge that there could be a similar imbalance in the prediction of

the subclasses of seen and unseen superclasses if they are mixed. This could be an open challenge in the CGFS task which is rarely discussed in the existing works. To alleviate this problem, some specific technical extensions may be designed, as discussed in (Xian et al., 2018; Oreshkin et al., 2020), which is out of our scope. We leave this scenario in our future work.

## C    APPENDIX FOR IMPLEMENTATION DETAILS

### C.1    THE SETUP OF THE COMPARED MODELS

For MoCo-v2 and MoCo-v2-ImageNet, following (Chen et al., 2020b), we set its queue size 65536, momentum 0.999 (0.99 for CIFAR-100), temperature 0.2, embedder $\mathcal{E}$ as MLP. For SwAV-ImageNet, following (Caron et al., 2020), its number of prototypes was 3000, embedder $\mathcal{E}$ as MLP. For ANCOR, following (Bukchin et al., 2021), we set its queue size 65536, momentum 0.999 (0.99 for CIFAR-100), temperature 0.2, embedder $\mathcal{E}$ as $fc$ and MLP, with the multi-class queue, and angular norm on. For GEORGE (Sohoni et al., 2020), we set its robust learning rate $1e^{-4}$, $\mathcal{E}$ as $fc$, (dim 16, 10 neighbors, 0 distance) for its dimension reduction step, the number of clusters per superclasses was grid-searched within $\{5, 10, 15, 20\}$. For SeLa, following (Asano et al., 2020), we set its $\lambda = 25$, $\mathcal{E}$ as $fc$, and number of clusters was grid-searched from 100 to 500 with step size 50. For Coarse+ and Fine+, we set $\mathcal{E}$ as MLP. For our SCGM, we set $\gamma = 0.5$, $\sigma^2 = 0.1$, and $\lambda = 25$ ($\lambda$ follows (Asano et al., 2020)), $\mathcal{E}$ as $fc$, the number of latent variables $r$ was grid-searched from 100 to 500 with step size 50.

**Remarks on embedder.** For the compared methods, adding an embedder follows the practice of the existing works such as MoCo-v2 (Chen et al., 2020b) and SimCLR (Chen et al., 2020a), which was also used in the different compared methods in (Bukchin et al., 2021). In these works, adding certain types of embedder (e.g., MLP) is empirically found helpful for improving performance. We have the same observation in our experiments, and think the effectiveness could come from (1) more capacity; and (2) better generalization in some cases. The latter is because after training, the embedder may be specific to the pre-training tasks, which enables more generalizable representations to be learned by the backbone networks. Thus dropping the embedder at testing stage may help alleviate overfitting to some extent. For comprehensiveness, we evaluated both of the cases when an embedder is added or not in our experiments, as described in Section 4.1.

### C.2    THE ARCHITECTURE OF SCGM-A

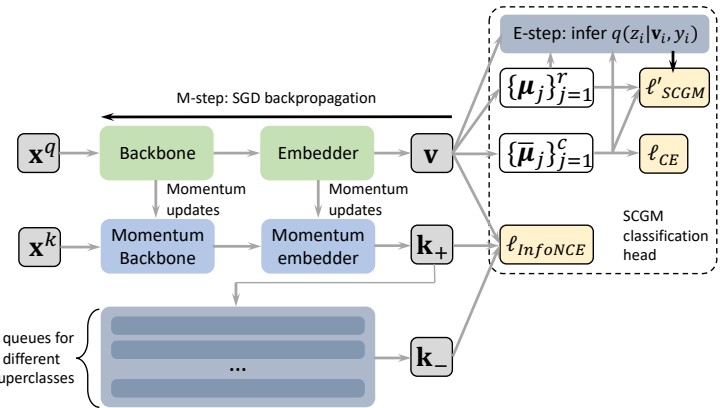

Figure 4: An illustration the architecture of SCGM-A

Fig. 4 illustrates the architecture of SCGM-A, which was implemented on a generic encoder (*e.g.*, ResNet-50) with momentum encoders. The query $\mathbf{v}$ and positive key $\mathbf{k}_+$ are computed from two random augmentations of the input images through the encoder and embedder, and their momentum updates. There is a superclass-wise dictionary maintains $c$ queues for $c$ different superclasses so that superclass-wise negative keys $\mathbf{k}_-$ can be generated for within-superclass contrastive learning, which facilitates preserving intra-class variation. At the SCGM classification head, $\mathbf{v}$, $\mathbf{k}_+$ and $\mathbf{k}_-$ are applied with an angular normalization, followed by the InfoNCE loss, where the angular normalization

is defined by (Bukchin et al., 2021)

$$|\mathbf{v}_i|_{\mathrm{A}} = \left( \frac{\mathbf{v}_i}{\|\mathbf{v}_i\|} - \frac{\bar{\boldsymbol{\mu}}_{y_i}}{\|\bar{\boldsymbol{\mu}}_{y_i}\|} \right) / \| \frac{\mathbf{v}_i}{\|\mathbf{v}_i\|} - \frac{\bar{\boldsymbol{\mu}}_{y_i}}{\|\bar{\boldsymbol{\mu}}_{y_i}\|} \| \tag{18}$$

which enables contrasts from an angular perspective, and induces better synergy with cross-entropy loss.

The top part of the SCGM classification head are similar to its counterpart using a generic encoder, where $\mathbf{v}$ interacts with superclass means $\{\bar{\boldsymbol{\mu}}_j\}_{j=1}^c$ and subclass means $\{\boldsymbol{\mu}_j\}_{j=1}^r$ for computing our SCGM loss $\ell_{\boldsymbol{\theta},\boldsymbol{\phi}}$ and a cross-entropy loss $\ell_{CE}$ as in Eq. (8). Thus the loss for training SCGM-A is

$$\ell(\mathcal{D}_{\text{train}}; \boldsymbol{\theta}, \boldsymbol{\phi}) = \ell_{CE}(\mathcal{D}_{\text{train}}; \boldsymbol{\theta}, \{\bar{\boldsymbol{\mu}}_j\}_{j=1}^c) + \ell_{\text{InfoNCE}}(|\mathcal{D}_{\text{train}}|_{\mathrm{A}}; \boldsymbol{\theta}) + \gamma \ell_{\boldsymbol{\phi},\boldsymbol{\theta}}(\mathcal{D}_{\text{train}}; \boldsymbol{\theta}, \boldsymbol{\phi}) \tag{19}$$

where the model parameters are learned by our proposed EM-based algorithm that alternates between the SGD backpropagation (M-step) for computing model parameters $\{\boldsymbol{\theta}, \boldsymbol{\phi}\}$ and the posterior inference (E-step) for computing subclass distributions.

### C.3  THE ARCHITECTURE OF DCCN

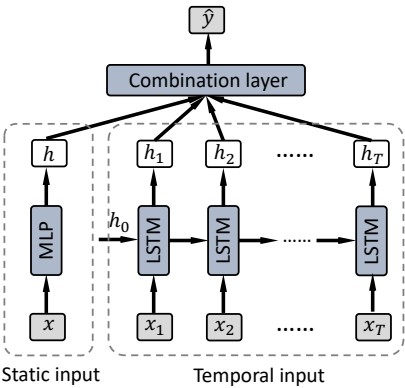

Figure 5: An illustration the architecture of DCCN

Fig. 5 illustrates the architecture of the Dual-Channel Combiner Network (DCCN). DCCN is effective to process heterogeneous medical records data that usually consists of static profiles and time series (*i.e.*, records) (Che et al., 2016). It has a static channel realized by MLPs to encode static features (*e.g.*, demographic information, infrequent blood test results, *etc.*) and a temporal channel realized by RNNs (*e.g.*, LSTM) to encode temporal feature (*e.g.*, blood flow, venous pressure, *etc.*). The hidden representations output by the two channels are concatenated (or using certain pooling method) and further projected to a compact embedding that is used for prediction by the classification head, *i.e.*, the combination layer in Fig. 5, which is realized by MLPs. Attention layer may be added before the combination layer for weighted combination of the hidden representations from different channels and different time steps.

## D  APPENDIX FOR MORE EXPERIMENTAL RESULTS

### D.1  EFFECTS OF MORE SHOTS

| Method | 1-shot | 5-shot | 10-shot | 20-shot |
|--------|--------|--------|---------|---------|
| Coarse+ | 37.44±0.12 | 50.51±0.31 | 57.35±0.33 | 63.23±0.25 |
| ANCOR | 45.14±0.12 | 59.60±0.35 | 65.14±0.28 | 68.87±0.26 |
| ANCOR-*fc* | 46.19±0.16 | 60.64±0.30 | 64.31±0.27 | 66.85±0.26 |
| SCGM-G | 48.74±0.15 | 60.39±0.29 | 63.42±0.31 | 65.67±0.25 |
| SCGM-A | **49.31±0.16** | **63.82±0.32** | **67.43±0.30** | **69.99±0.25** |

Table 7: Effects of more shots (all-way)

Table 7 and 8 summarize the performance of SCGM-G and SCGM-A, with the strongest baseline ANCOR and ANCOR-*fc*, together with the coarse baseline Coarse+, on Living17 dataset. From

| Method | 1-shot | 5-shot | 10-shot | 20-shot |
|---|---|---|---|---|
| Coarse+ | 79.29±0.65 | 92.99±0.41 | 94.33±0.36 | 95.27±0.31 |
| ANCOR | 89.23±0.55 | **94.44±0.36** | **95.33±0.31** | **95.90±0.27** |
| ANCOR-*fc* | 90.41±0.57 | 93.84±0.36 | 94.52±0.32 | 95.09±0.29 |
| SCGM-G | 89.72±0.54 | 94.38±0.37 | 95.09±0.35 | 95.52±0.31 |
| SCGM-A | **90.97±0.55** | 94.41±0.34 | 95.11±0.30 | 95.43±0.28 |

Table 8: Effects of more shots (5-way)

Table 7, we observe SCGM-A continuously improves performance as more shots are added, consistently outperforms other models, which validate the effectiveness of SCGM framework with varying number of shots. SCGM-G's performance becomes comparable to ANCOR(-*fc*) as more shots are added, due to its simpler architecture, which however is more computationally efficient. Thus it provides a good balance between performance and efficiency. From Table 8, we observe ANCOR and SCGM perform similarly. This is because the task is close to the regular superclass classification, and its performance limit can be easily reached by adding a few more shots. This can be seen from Coarse+'s results, which become close to other models when there are 20 shots in the support set.

## D.2 IMPACTS OF THE NUMBER OF SUBCLASSES

| Setting | $r$=50 | $r$=100 | $r$=150 | $r$=200 | $r$=250 | $r$=300 | $r$=400 | $r$=500 |
|---|---|---|---|---|---|---|---|---|
| 5-way | 89.45±0.60 | 89.72±0.54 | 88.29±0.57 | 88.26±0.55 | 88.05±0.56 | 87.88±0.56 | 86.82±0.56 | 86.35±0.58 |
| All-way | 42.98±0.14 | 48.74±0.15 | 46.21±0.15 | 47.81±0.15 | 47.55±0.16 | 46.91±0.16 | 45.23±0.15 | 44.63±0.15 |

Table 9: Impacts of the number of latent variables $r$ on the performance of SCGM-G

Table 9 presents the results of SCGM-G on Living17 by varying the number of latent variables $r$ from 50 to 500. As can be seen, the performance of SCGM-G is generally stable and it consistently outperforms Coarse(+) (Table 1) w.r.t. different $r$. By looking into the details, we observe too small (*e.g.*, 50) or too big (*e.g.*, 500) $r$ may degenerate the performance in the all-way case, which may be caused by slight underfitting and overfitting, respectively. In the 5-way case, small $r$ does not impact performance obviously, this is because the 5-way case is close to evaluate coarse classification, which does not need exact inference of subclasses. Thus a small number of $r$ is fine (which may not distinguish all possible subclasses).

## D.3 IMPACTS OF THE VARIANCE

| Setting | $\sigma^2$=0.05 | $\sigma^2$=0.1 | $\sigma^2$=0.15 | $\sigma^2$=0.2 | $\sigma^2$=0.25 | $\sigma^2$=0.3 |
|---|---|---|---|---|---|---|
| 5-way | 90.49±0.56 | 89.72±0.54 | 90.09±0.57 | 90.88±0.55 | 90.56±0.59 | 90.69±0.59 |
| All-way | 46.47±0.15 | 48.74±0.15 | 47.48±0.15 | 48.03±0.16 | 42.21±0.13 | 38.38±0.13 |

Table 10: Impacts of the variance parameter $\sigma^2$ on the performance of SCGM-G

Table 10 presents the results of SCGM-G on Living17 by varying the variance $\sigma^2$ of subclasses, which resembles the temperature scaling in Eq. (7). As described in Sec. 3.1, in our experiments, we set $\bar{\sigma}^2 = 1$ and tune $\sigma^2$ to adjust the relativity between super- and subclasses, which we found is empirically effective. From Table 10, we observe $\sigma^2$ is better to be selected from 0.1 to 0.2, and a too small (*e.g.*, 0.05) or too big (*e.g.*, 0.3) value may degenerate the performance in the all-way case. This observation is similar to the evaluation of the temperature scaling parameter in MoCo-v2 (Chen et al., 2020b). In the 5-way case, the performance is relatively stable w.r.t. different values of $\sigma^2$. Similar to the observation in Appendix D.2, this is because the 5-way case is close to evaluate coarse classification, which is less impacted by the learning of subclasses.

## D.4 CONVERGENCE ANALYSIS

Fig. 6 presents the training loss $\ell(\mathcal{D}_{\text{train}}; \boldsymbol{\theta}, \boldsymbol{\phi})$ in Eq. (8) of SCGM-G, and $\ell(\mathcal{D}_{\text{train}}; \boldsymbol{\theta}, \boldsymbol{\phi})$ in Eq. (19) of SCGM-A, during the training process on the four BREEDS datasets. The training alternates between E-step and M-step, where E-step runs every 5 epochs. As can be seen, although there are some periodic fluctuations, the loss function values generally decrease and converge to a small value across different datasets. The periodic fluctuations are not synchronized with the alternate E-step and M-step since its period is not 5 epoch as can be observed. In fact, the fluctuations come from the

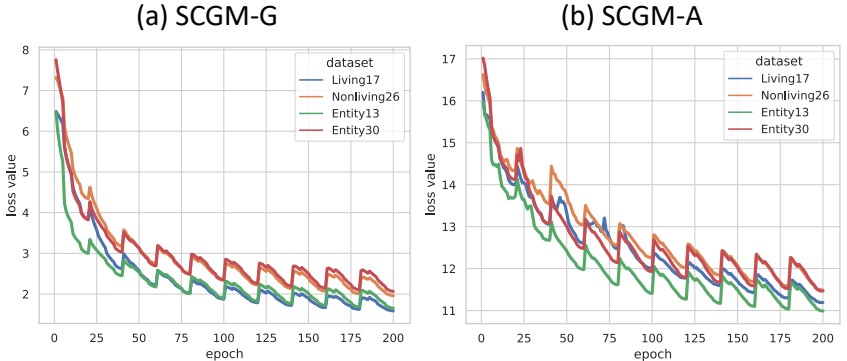

Figure 6: The training loss w.r.t. the number of epochs of SCGM on BREEDS datasets

cosine annealing with warm restarts schedule for the learning rate (as discussed in Sec. 4.1), which has 20 epochs per cycle. This is consistent with Fig. 6. In practice, we observed both E-step and M-step can decrease the loss function values since both steps are theoretically optimizing the same loss function. There is a rounding step (as discussed in Sec. 3.2) for generating discrete code from the posterior, which follows the E-step. We found this rounding step is beneficial, since without it, too small values may be generated in the posterior inference, which is unstable in computation. Generally, the current setup provides a stable training framework and we don't need to interfere in the program during the training process on all of our experimental datasets.

## D.5 VISUALIZATION OF EMBEDDING

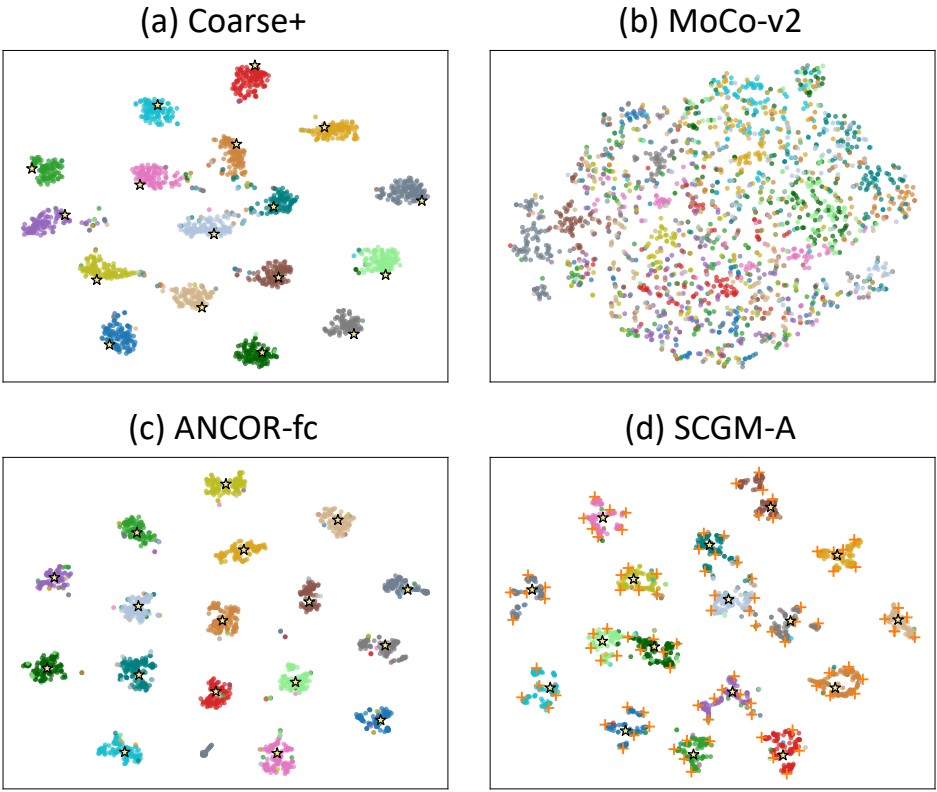

Figure 7: The tSNE visualization of the embeddings learned by different methods. Star represents superclass means. "+" marker represents subclass means.

In addition to the visualization results of ANCOR and SCGM-G provided in Fig. 2, Fig. 7 presents the tSNE visualization of the embeddings of several most relevant methods in comparison using Living17 dataset, including Coarse+, MoCo-v2, ANCOR-*fc* and SCGM-A. In the figure, colors mark different superclasses, "+" markers are the learned subclass means by SCGM-G, and stars represent superclass means. From Fig. 7(a)(c), we can observe the embeddings of both ANCOR and ANCOR-*fc* resemble Coarse+. Although ANCOR-*fc* slightly distinguish several small groups of embeddings in some superclasses, their intra-class variation is suboptimal. From Fig. 7(b), MoCo-v2 has the ability to detect superclasses to some extent, but cannot fill the gap of superclass supervision, by comparing with Coarse+, in this dataset. The reason may be the Living17 dataset is not sufficiently large and comprehensive for MoCo-v2 to learn embeddings that are comparable to supervised methods. In contrast to them, SCGM-A (and SCGM-G in Fig. 2) explicitly detects subclasses, with proper position of subclass means and superclass means. It also associates every subclass to its corresponding superclass, which establishes a hierarchical class structure. This explains the effectiveness of SCGM in cross-granularity learning tasks.

