# OpenReview forum: "Superclass-Conditional Gaussian Mixture Model For Learning Fine-Grained Embeddings"
_ICLR.cc/2022/Conference — ICLR 2022 Spotlight_

### Official Review · Reviewer_w8nz · 2021-10-31

**Correctness:** 4
**Technical Novelty And Significance:** 3
**Empirical Novelty And Significance:** 3
**Recommendation:** 6
**Confidence:** 2

**Main Review:**

Strengths:
1. The paper is well-motivated and addresses the important problem of few-shot multi-granularity adaptation.
2. The paper includes comprehensive evaluation across various datasets and demonstrate the effectiveness and stability of the proposed approach.
3. Visualization confirms the proposed approach learns the superclass-subclass structure.

Weaknesses:
The main focus of the proposed approach is learning embeddings, but the evaluations are focused on classification. If the goal is classification, why not directly build a classification model? If the goal is learning embeddings, it would be better to put more focus on embedding based evaluation and clarify the relationship between embedding and classification performance.

**Summary Of The Paper:**

The paper introduces a setup with the goal to adapt from a coarse pretrained model to unseen fine-grained labels. This problem is formulated as a superclass-subclass latent model and learned with maximum likelihood via expectation-maximization. The proposed approach, super-class conditional Gaussian mixture (SCGM) model, defines a hierarchical Gaussian distribution on the class hierarchy that models both the superclasses and subclasses. SCGM is evaluated on 6 image-net related datasets and a a real Dialysis-Event dataset collected by hospitals, and demonstrates competitive results under two evaluation setups: generalizing to seen and unseen superclass.

**Summary Of The Review:**

The paper introduces the interesting problem of few-shot multi-granularity adaptation through hierarchical Gaussian mixture and demonstrates its effectiveness across multiple datasets. The relation between the goal (embedding learning) and evaluation (classification accuracy) is not very clear.

---

> ### Author Response · Authors · 2021-11-16
> **Response to the comments from the reviewer**
>
> Thank you so much for the constructive feedback. We sincerely appreciate your valuable suggestions and questions. The following are our responses. We appreciate if the reviewer could let us know if there are any further questions.
>
> Q1. The relation between the goal (embedding learning) and evaluation (classification accuracy) is not very clear.
>
> Thanks for pointing it out. We are now aware of the confusion that may be caused by mentioning "embedding" in the title.
>
> In this work, we focus on solving the cross-granularity few-shot (CGFS) learning problem (as described in the second paragraph of Section 1 "Introduction"), which directly connects to the problem of learning "good" embeddings for effective adaptation according to the framework of some recent works such as
> * Tian, et al. "Rethinking few-shot image classification: a good embedding is all you need?." In ECCV, 2020.
>
> and some other works as discussed in the "Few-shot learning" section of Section 2 "Related work". In the CGFS task, a "fine-grained" embedding that can distinguish subclasses (by only learning from superclass labels) is considered as a "good" embedding in our opinion. (Only using superclass labels for training is because the (fine-grained) subclass labels are usually more difficult to obtain than the (coarse) superclass labels.)
>
> Therefore, our goal of solving the CGFS problem is realized by learning "fine-grained" embedding using only superclass labels, and our evaluations are all about the performance of our model on the CGFS problem. From this perspective, we think the goal and the evaluation are consistent, and our original intention of using "fine-grained embedding" in the title is due to its unique importance in solving the problem.
>
> Q2. If the goal is classification, why not directly build a classification model?
>
> Since our goal is to solve the CGFS problem rather than a classification problem in general, building a classification model may not be suitable. This is because the tasks of the pre-training stage and testing stage are different. The former is about superclasses (coarse labels), and the latter are about different subclasses (fine-grained labels). The two tasks are different in the semantic meaning and the number of the two disjoint sets of classes. Therefore, a classification head trained on superclasses may not be usable on different testing tasks on subclasses, which requires the adaptation of the embedding model (the architecture before the classification head) because the embedding model is more agnostic to different tasks and generalizable.
>
> Q3. If the goal is learning embeddings, it would be better to put more focus on embedding based evaluation and clarify the relationship between embedding and classification performance.
>
> Different from some general embedding methods, such as some unsupervised and self-supervised methods, which aim to encode the data structure in general, our goal is to train an embedding model that is able to learn the hierarchical structure of classes (with supervision only from superclasses), which is important for filling the gap of granularity between the training and testing scenarios (i.e., solving the CGFS problem). Therefore, to evaluate whether the embeddings effectively encode such a structure and distinguish different subclasses, we focus on the evaluation of the cross-granularity adaptation capability of the embedding models (with two evaluation cases as described in Section 4.2 and a case study in Section 4.3). We think this evaluation aligns better with our task of learning "fine-grained" embeddings than some other downstream tasks for evaluating the general embedding methods.

---

> > ### Comment · Reviewer_w8nz · 2021-11-19
> > **Thanks for the response**
> >
> > I'd like to thank the authors for clarifying my confusions in the response!

---

> > > ### Author Response · Authors · 2021-11-19
> > > **Thank you**
> > >
> > > Thank you for reviewing our response, and we are happy that our response helps clarify your confusions. We also appreciate if the reviewer could let us know if there are any remaining questions or concerns that make you feel we haven't sufficiently justified our work for a higher score. We appreciate if the reviewer could let us know where we can refine or what concerns you still have, so that we can take the opportunity to clarify them and improve our work. Thank you!

---

### Official Review · Reviewer_FxzL · 2021-11-02

**Correctness:** 3
**Technical Novelty And Significance:** 3
**Empirical Novelty And Significance:** 3
**Recommendation:** 8
**Confidence:** 4

**Main Review:**

Strengths of the paper include the following:

(1) The problem addressed is an important and understudied one.  While previous work has identified the CGFS problem (though not necessarily called it by that name), methods to address it in a technically principles way remain in their infancy.

(2) The method proposed both naturally follows from the existing treatment of the problem in the literature and presents a novel approach to address it.  The generative modeling framework presented by the authors is well-motivated, described with a reasonably high degree of precision, and its relationship to other approaches (e.g. GEORGE, contrastive representation learning) is analyzed in a compelling manner.

(3) Empirical results are generally compelling, and comparison between the proposed method and existing work (a) takes into account relatively current approaches to the problem and (b) evaluates performance along several important axes, including both classification performance and systems-level performance.

Weaknesses of the paper include the following:

(1) Though the presentation and analysis of the proposed method is reasonable, comparisons to other methods are at some points overstated in the opinion of this reviewer.  Several statements (mentioned in the “detailed comments” below) should be either refined, repositioned, or omitted to accurately reflect the contribution of this work.

(2) Empirical performance gains with respect to ANCOR in particular are not always convincing.  There exist a number of cases where the two are very close, and it is not clear how evaluations of superiority/non-inferiority between different methods was performed.  The authors should make their procedure for computing confidence intervals and determining superiority/noninferiority clear, and should ideally provide appropriate statistical tests/analysis for these claims.

(3) It would be desirable to have some analysis of theoretical guarantees about worst-case subclass performance.  The paper does not provide any such analysis in its current state.  It also appears (at least as far as this reviewer understands it) that the current formulation imposes a constraint that subclasses be of equal size.  If true, this should be discussed in substantially more detail, and would make the worst-case subclass performance analysis even more important.

**Summary Of The Paper:**

The paper presents SCGM, a new technique for solving the Cross-Granularity Few-Shot learning (CGFS) problem.  CGFS is defined as the problem of adapting a classification model trained on coarse (“superclass”) labels to perform well on fine-grained labels, which consist of multiple “subclass” labels within each superclass.  The paper presents a new generative modeling approach that enables end-to-end classifier training in a manner that (a) incorporates information about superclass-subclass hierarchies (b) does not rely on explicit subclass enumeration (c) provides improved empirical performance on CGFS and (d) provides systems benefits with respect to existing baselines.  The authors provide detailed description of their approach and characterize its relationship with existing methods, and present empirical results that support their methodological choices.

**Summary Of The Review:**

Abstract:

”It is crucial to fields where fine-grained labeling (e.g., breeds) requires strong domain expertise thus is prohibitive, such as medicine, but predicting them is desirable.” --> I found this sentence confusing, consider rephrasing.

Introduction:

“During the treatments…task” --> I understand the general point here, but the example seems a bit strange.  For instance, if I could predict an event before performing dialysis, would I not consider delaying the procedure?  If I delay the procedure, the events never happen.  If it's useful to be able to say "we predict event type A vs. event type B" to help physicians assess the risk of performing the procedure, that would make sense...but that doesn't seem to be how it's presented here.  Consider reframing this to make it reflect the situation you're trying to describe more precisely (or consider providing a slightly different example).  This is a minor issue, but the framing could throw some readers off.

“The data constitute a support set” --> In the sense that they provide support for the different fine-grained annotation schema?  Or for training/fine-tuning?  Or for both?  Additional precision would help here.

“Unseen siblings of the superclasses” --> What is meant by this phrase?

“Subclasses may arbitrarily spread within each superclass” --> Maybe point out "and in real datasets, it is often the case that they are unevenly represented in a way that causes subclass performance gaps."

“However, their approach is confined to images” --> Why is this true?  Certainly the referenced work focuses on images, but it is not obvious to me why it could not be directly extended to the type of data in medical records.  I am also not sure I agree that there are not reasonable/standard approaches for obtaining positive/negative pairs on other types of data.  Even if this were true, however, I would argue that it's not a downfall of Bukchin et al.  Recommend rephrasing or cutting this piece.

“Also, since contrastive learning does not model subclasses…Sec .4).” --> At this point in the paper, it's not clear at this point how you support these statements in Sec. 4, and the claims may seem unwarranted.  Recommend either deferring this discussion or providing some quantitative previews of your results here to mitigate this.

“It explicitly represents the unobserved subclasses as latent variables…” --> It may be worth clarifying here that you do not assume knowledge of the subclass identities.

Related Work:
“…cannot trivially be extended to non-image data…” --> See comment above, I don’t think this is a great argument.  Recommend cutting or rephrasing.

“…cannot leverage coarse training to guide their pseudo-labeling…” --> This claim is unclear. Coarse training explicitly does guide the pseudo-labeling in e.g. GEORGE (Sohoni et al. 2020), as clusters w/in the coarsely trained representation define the pseudolabels.  The argument about end-to-end optimization of the pseudo-labels/model parameters seems reasonable, but not sure this statement is accurate.  Maybe rephrase to indicate that a two-step approach could lead to suboptimal psuedolabeling rather than arguing that the coarse representation doesn’t contribute at all in these two-step methods.

“These methods are unsupervised, and cannot be used…” --> Why does the unsupervised nature of these approaches mean they are not capable of application so this problem?  I found this part unclear…I suspect it may be a writing issue rather than a technical issue, but clarification would be appreciated.

Superclass-conditional Gaussian Mixture Model:

“unseen superclass” --> What does an unseen superclass mean?  Is it an unseen subgroup that spans existing superclasses?  Or an OOD example?  Need a more precise description of what this means, recommend providing mathematical definition.

“Unlike some previous works (Sohoni et al., 2020), which allocated a predefined number of subclasses to every superclass…” --> This isn't true, as far as I understand it.  Sohoni et al. perform a search over the number of subclasses w/in each superclass and optimize an unsupervised cluster quality metric.  One could argue that fewer searches are required in the method this paper proposes – because all of the searching over number of subclasses is baked into the r parameter – but I don’t think that argument is particularly compelling from a practical standpoint.  I would suggest cutting or softening this claim.

Figure 1 --> Appears to be a typo in the caption, all of the subclass means have the same subscript (“1”)

“…and tune \sigma to adjust the relativity between super- and sub-classes” --> What range does this usually fall in?  Worth reporting this.

“transport polytope” --> Many readers may not be familiar with this.  Recommend providing a brief description/background on this in the Appendix (and reference this in the text)

“It is noteworthy…” --> This is an interesting set of observations.  Nit: typo in “contrasitive”
Experiments:

“visual granularity” --> What exactly does this mean?  A comment would be helpful.

“20/6/8 splits” --> So the superclasses are disjoint in train/test/val?  That seems a little confusing.  Suggest clarifying

“which adds an embedder” --> What exactly does this embedder do/why is it added? More representational capacity?  More detail would be welcome.

“dim” --> “dimension”

Table 1: Was there a statistical test done here to support improvements over ANCOR?  Seems to be overlapping confidence intervals in a couple of cases.  Please provide detail on how confidence intervals were computed and what statistical approach was used to determine superiority/non-inferiority.  Also, what's similar about the cases where the method outperforms and doesn't outperform ANCOR?  Is there any additional explanation that can be given here?

“baselines on contrastive learning cannot be applied…" --> see above comments, I don’t think this is a particularly strong argument.  I would cut it, or at least acknowledge that this is not a fundamental limitation of the contrastive approaches.  Augmentation schemes certainly exist for non-image data types.  The strongest response would run these baselines using data augmentation approaches for these other modalities.

Appendix A:

Eq. (10) --> I am not following exactly how Jensen’s inequality results in the move from line 1 to line 2 here.  I could be missing something, but it would certainly help if the authors provided more explicit detail in the intermediate steps (e.g. laying out the terms in the inequality clearly – i.e. a_i and x_i here https://en.wikipedia.org/wiki/Jensen%27s_inequality).

“the second step in Eq. (11)…” --> I think this is a typo…should be Eq. (10) referenced here?

“\lambda \ge 1” --> Am I correct in understanding that there is strict equality for \lambda = 1?

“equal partition constraint” --> If I understand correctly, this statement deserves more discussion.  Does this imply that the constrained transport polytope formulation here effectively assumes that subclasses are of equal size?  This would appear to be consistent with Eq. (6), and if this is indeed true this would represent a substantial limitation of the method that should be further discussed.  On this note, what is the distribution of subclass sizes in the experiments run in this paper?  Are the subclasses all of similar size, or does it vary?

Eq. (13) --> the last equality/definition here seems a bit strange.  You remove the 1 constant, and define the term without the 1 as equal to the term without the 1?  Why is this true/necessary?

Appendix B:

“…robust learning rate…” --> Could be worth searching over robust learning rate for this baseline.

Appendix C:

Table 8 --> Suggest adding bolding to this table as well to maintain consistency with other tables.

---

> ### Author Response · Authors · 2021-11-16
> **Response to General Questions (Part 1)**
>
> Thank you so much for the constructive feedback. We sincerely appreciate your valuable suggestions and questions. The following are our responses.
>
> Q1. Several statements (mentioned in the "detailed comments") should be either refined, repositioned, or omitted to accurately reflect the contribution of this work.
>
> This question is related to the questions from Q5 to Q36 in the following. The responses are summarized to each of the questions.
>
> Q2. It is not clear how evaluations of superiority/non-inferiority between different methods was performed. The authors should make their procedure for computing confidence intervals and determining superiority/noninferiority clear, and should ideally provide appropriate statistical tests/analysis for these claims.
>
> This question is related to questions Q26, Q27, Q28 is the following. The response is summarized in the response to Q26, Q27, Q28.
>
> Q3. It would be desirable to have some analysis of theoretical guarantees about worst-case subclass performance. The paper does not provide any such analysis in its current state. It also appears (at least as far as this reviewer understands it) that the current formulation imposes a constraint that subclasses be of equal size. If true, this should be discussed in substantially more detail, and would make the worst-case subclass performance analysis even more important.
>
> This question is related to question Q33 is the following. The response is summarized in the response to Q33.

---

> > ### Comment · Reviewer_FxzL · 2021-11-22
> > **Thanks for the response!**
> >
> > The authors have done an excellent and thorough job addressing my questions.  I very much appreciate their time and effort in doing so, and I have increased my score as a result.
> >
> > Some brief thoughts:
> >
> > (1) Q7: Adding text in the manuscript clarifying the explanation the authors provide here about exactly what is meant by the “support set” would be helpful to the reader.
> >
> > (2) Q22 - Q24: Thanks for the explanations; recommend adding similar explanations to the manuscript.  Q24 answer in particular could go in an appendix.
> >
> > (3) Q26: Do your “random episodes” refer to bootstrapping on the test set?  I think this is true, but worth clarifying.
> >
> > (4) Q28: This helps.  Suggest adding a comment like this to the manuscript.
> >
> > (5) Q33: Thanks for the detailed response — adding this section about the equal partition constraint is helpful.

---

> > > ### Author Response · Authors · 2021-11-23
> > > **Response to the suggestions from the reviewer**
> > >
> > > Thank you for reviewing our response, and we are happy that our response helps with the questions. We also appreciate your recognition of our work! The following are our responses to your suggestions.
> > >
> > > (1) Q7: Adding text in the manuscript clarifying the explanation the authors provide here about exactly what is meant by the "support set" would be helpful to the reader.
> > >
> > > Thanks for the suggestion. We revised the beginning part of the second paragraph in Section 1 to make the explanation more clear. Now it becomes "To fill the gap of granularity between the training and testing scenarios, a practical way is to collect a few new records for a patient, with their fine-grained annotations. These data constitute a support set for fine-tuning a pre-trained model to the specific data distribution induced by the annotations of the target patient, for whom the adapted model is used for future predictions.".
> > >
> > > (2) Q22 - Q24: Thanks for the explanations; recommend adding similar explanations to the manuscript. Q24 answer in particular could go in an appendix.
> > >
> > > Q22: We revised the corresponding part in the second paragraph of Section 4 to make "visual granularity" more clear. Now it becomes "visual granularity (i.e., the granularity of images for distinguishing classes)".
> > >
> > > Q23: We added "(disjoint)" in the description of the third dataset (the second paragraph of Section 4). Now it becomes "... for (disjoint) train/val/test sets.". In Section 4.2 ("Evaluation case (2)"), we described "Since tieredImageNet has distinct train/val/test sets, it was used to evaluate the case when the testing subclasses is out of any training superclass. All models were trained with the superclasses of the train set, and tested on the subclass labels of the test set." to make the setup clear.
> > >
> > > Q24: According to the suggestion, we added a section in Appendix C.1 ("Remarks on embedder") to include our response to Q24.
> > >
> > > (3) Q26: Do your "random episodes" refer to bootstrapping on the test set? I think this is true, but worth clarifying.
> > >
> > > Following (Tian et al., 2020, Bukchin et al., 2021), the "random episodes" refer to sample a random subset of all subclasses to constitute the classes of the support set and query set in each episode. The support samples and query samples are then randomly drawn from the sampled subset of subclasses. To help make it clear, we described this in the sentence below Table 3 as "The test classes of each episode were a random subset of all subclasses (or all of them for all-way tests).".
> > >
> > > (4) Q28: This helps. Suggest adding a comment like this to the manuscript.
> > >
> > > According to the suggestion, we added "The results suggest SCGM could perform better when more subclasses are from the same superclass." before the last two sentences in Section 4.2 ("Evaluation case (1)").
> > >
> > > (5) Q33: Thanks for the detailed response - adding this section about the equal partition constraint is helpful.
> > >
> > > Thank you so much for this suggestion. It helps us to include the discussion in Appendix B.1.

---

> ### Author Response · Authors · 2021-11-16
> **Response to General Questions (Part 2)**
>
> Q4. Why the method in (Bukchin et al., 2021) cannot be directly applied to the type of data in medical records? Are there any standard approaches for obtaining positive/negative pairs on different types of data?
> (This is a summary of questions Q10, Q13, Q29 in the following)
>
> The contrastive learning based method in (Bukchin et al., 2021), which was built upon MoCo (He et al., 2020), generates a positive pair by augmenting another "view" of an original image. The augmentation is done by modifying the original image through methods including random crop, random color jittering, random horizontal flip, random grayscale conversion, and Gaussian blur (He et al., 2020, Chen et al. 2020b). The method SimCLR has similar operations (Chen et al. 2020a). The negative pair is generated if two images do not originate from the same image. These augmentation methods are specific to images, and cannot be trivially applied to other types of data. In our understanding, the data augmentation step requires some human prior of a specific data type on how two samples can be regarded as a positive pair (e.g., two random "views" of an image have similar semantics).
>
> When extending these contrastive learning methods to other types of data, specific augmentation methods may be designed. For example, on graphs, (You, et al. "Graph contrastive learning with augmentations." In NeurIPS, 2020.) augmented data by methods including node dropping, edge perturbation, attribute masking, and subgraph sampling, which are specific to graphs. On time series, (Tonekaboni, et al. "Unsupervised representation learning for time series with temporal neighborhood coding." In ICLR, 2021.) sampled positive pairs within certain temporal neighborhoods, (Eldele, et al. "Time-series representation learning via temporal and contextual contrasting." In IJCAI, 2021.) used random permutation of segments of time series with jittering. On healthcare records, (Yeche, et al. "Neighborhood Contrastive Learning Applied to Online Patient Monitoring." In ICML, 2021.) used random dropout, random mask of variables, and random Gaussian noise to generate positive pairs.
>
> We agree with the reviewer that augmentation methods exist for non-image data. Meanwhile, we think these different data augmentation methods are mostly under exploration, and are not as widely used as those methods on image data. So it may be hard to say these works provide a standard approach.
>
> Therefore, in the previous draft, we wrote "the approach of (Bukchin et al., 2021) cannot be readily used for medical records" (Section 1), and "cannot be trivially extended to non-image data due to the absence of a general method for data augmentation" (Section 2). Here, by using "general", we mean a method that can be used for different types of data and different model architectures, which seems unavailable, to the best of our knowledge. It may be possible to extend the method of (Bukchin et al., 2021) using the augmentation methods above, such as (Yeche, et al. 2021), but it may take some (non-trivial) efforts to accommodate the distinction between the network architectures and training losses of these two works.
>
> We also agree with the reviewer that this is not a fundamental limitation of a constrastive learning based method. By mentioning it, we want to demonstrate a difference between our proposed model and the one in (Bukchin et al., 2021). That is, our proposed model is more flexible to different applications because it does not require searching for a specific augmentation method for a specific type of data, considering an augmentation method may be under exploration.
>
> According to the reviewer's suggestion, we revised our description in the draft. In Section 1, we cut "their approach is confined to images" to acknowledge this is not a fundamental limitation of the method (according to Q10). In Section 2, we rephrased "... cannot be trivially extended to ..." by "require (non-trivial) searches and trials for a suitable data augmentation method, which may be unavailable for some non-image data." (according to Q13) In Section 4.3, we cut "baselines on contrastive learning cannot be applied" as suggested to avoid confusion (according to Q29).

---

> ### Author Response · Authors · 2021-11-16
> **Response to the comments on Abstract**
>
> Q5. "It is crucial to fields ... predicting them is desirable." I found this sentence confusing, consider rephrasing.
>
> We rephrased this sentence in the revised draft. Now it becomes "It is crucial to fields for which fine-grained labeling (e.g., breeds of animals) is prohibitively costly, but predicting the fine-grained classes is desirable, such as medicine."

---

> ### Author Response · Authors · 2021-11-16
> **Response to the comments on Introduction**
>
> \* We revised the last paragraph of Section 1 to provide a bullet summary of our work according to the suggestion of Reviewer 1.
>
> Q6. "During the treatments ... task" Consider reframing this to make it reflect the situation you're trying to describe more precisely.
>
> Please let us clarify the scenario briefly. Because the binary labels only mark the incidence of an event (without its types), a model trained with these labels are expected to predict whether an event will happen. The prediction will help doctors to decide whether to perform the hemodialysis or delay it to another date when the patient gets better to reduce the risk. Compared to only predict the incidence of events, if a model can predict the types of events, then it may be useful for doctors to make a precise diagnosis, better evaluate the risk of a dialysis, and decide whether to delay the dialysis or perform it (with certain precautions). According to the reviewer's suggestion, we rephrased the description of this part in the first paragraph of Section 1. We appreciate if the reviewer could clarify it if there is any misunderstanding.
>
> Q7. "These data constitute a support set" In the sense that they provide support for the different fine-grained annotation schema? Or for training/fine-tuning? Or for both?
>
> The support set is used for finetuning a model that was pre-trained on the superclasses. Because the few-shot data in the support set are annotated with some subclasses, the main purpose of using the support set is to enable the adaptation of the pre-trained model to the specific data distribution induced by the subclasses.
>
> Q8. "Unseen siblings of the superclasses" What is meant by this phrase?
>
> The siblings mean other superclasses at the same hierarchical level (in the taxonomy, or the hierarchical tree of classes) of the superclasses used for model pre-training. These siblings are unseen because they are not observed at the pre-training stage. However, it is likely that their descendants (i.e., their subclasses) appear in the support set of some testing tasks. We rephrased this phrase in the draft. Now it becomes "other superclasses that are unobserved during pre-training".
>
> Q9. "Subclasses may arbitrarily spread within each superclass" Maybe point out "and in real datasets, it is often the case that they are unevenly represented in a way that causes subclass performance gaps."
>
> We understand this suggestion. It helps clarify a challenging case when the sizes of subclasses are very different, which may prevent accurate detection of some rare subclasses. If detecting those small subclasses is important in some applications, as discussed in (Sohoni et al., 2020), the method of (Sohoni et al., 2020) is a useful choice.
>
> We pointed it in the draft. Now it becomes "Subclasses may arbitrarily and unevenly spread within each superclass". Because our goal of solving CGFS problem is different from detecting those small subclasses (i.e., worst-case subclasses, which may require some specific designs of method), our description in this context focuses on the key challenge to the CGFS problem, i.e., preserve intra-class variation during training, so that there is less distraction.
>
> Q10. "However, their approach is confined to images" Recommend rephrasing or cutting this piece.
>
> We cut this piece in the draft. The response to this question is summarized in the response to General Question Q4.
>
> Q11. "Also, since contrastive learning does not model subclasses ... Sec. 4)" Recommend either deferring this discussion or providing some quantitative previews of your results here to mitigate this.
>
> We agree that descriptions without preview of the quantitative results may not strongly support the conclusion in this sentence. Due to the space limitation that may not be enough for presenting quantitative results, in the revised draft, please allow us to soften the description by replacing "their solution is suboptimal" with "their solution could be suboptimal", and referring to the results by adding "(evaluated in Sec. 4)".
>
> Q12. "It explicitly represents the unobserved subclasses as latent variables ..." It may be worth clarifying here that you do not assume knowledge of the subclass identities.
>
> We rephrased this sentence in the draft. Now it becomes "explicitly represents the unobserved subclasses by latent variables, without assuming their identities.".

---

> ### Author Response · Authors · 2021-11-16
> **Response to the comments on Related Work**
>
> Q13. "... cannot trivially be extended to non-image data ..." Recommend cutting or rephrasing.
>
> We rephrased the description. The response to this question is summarized in the response to General Question Q4.
>
> Q14. "... cannot leverage coarse training to guide their pseudo-labeling ..." Maybe rephrase to indicate that a two-step approach could lead to suboptimal psuedolabeling rather than arguing that the coarse representation doesn't contribute at all in these two-step methods.
>
> Thanks for pointing it out. We are now aware that the phrase of this sentence may not be rigorous. We changed it in the draft according to the suggestion. Now it becomes "the latter two methods could lead to suboptimal pseudo-labeling, and misleading labels could confuse the downstream step."
>
> Q15. "These methods are unsupervised, and cannot be used ..." Why does the unsupervised nature of these approaches mean they are not capable of application so this problem? I suspect it may be a writing issue rather than a technical issue, but clarification would be appreciated.
>
> Yes, it is because of writing. These unsupervised clustering methods only infer clusters agglomeratively or divisively, but do not learn data embeddings. So they cannot provide an embedding model that could be adapted to other tasks. Thus these methods cannot be directly applied in the few-shot learning scenario. We revised this sentence in the draft. Now it becomes "These unsupervised methods only infer clusters, but do not pre-train embedding models for task adaptation."

---

> ### Author Response · Authors · 2021-11-16
> **Response to the comments on Section 3**
>
> Q16. "unseen superclass" What does an unseen superclass mean?
>
> Similar to the response to Q8, unseen superclasses are superclasses that have not been used for model pre-training, but their subclasses may appear in the support set of some testing tasks. We changed the corresponding sentence in the first paragraph in Section 3 to "we also explored the case when the subclasses belong to superclasses that are not used for pre-training".
>
> Q17. "Unlike some previous works (Sohoni et al., 2020), which allocated a predefined number of subclasses to every superclass" Sohoni et al. performed a search over the number of subclasses w/in each superclass and optimize an unsupervised cluster quality metric.
>
> Yes, the method in (Sohoni et al., 2020) searches the number of subclasses within each superclass that yields the highest Silhouette score. It seems to be a kind of grid search over a certain range of numbers using a quality metric. As mentioned in the review, this search may take some time which may depend on the number of superclasses. It is to some extent because of the clustering method used for each superclass assumes an input number of subclasses, which should be allocated, either by a user or a searching algorithm. We agree the searching algorithm saves some efforts for tuning the number, but it assumes a range of the number of subclasses in each superclass, which is a kind of prior (but is weaker than without searching). In contrast, our proposed model assumes a total number of subclasses in all superclasses, so we think it helps alleviate the assumption one step further because the subclass partitions will be learned. As such, the model may be more generalizable with a weaker prior. According to the suggestion, we soften the sentence in the draft. Now it becomes "Unlike some previous works (Sohoni et al., 2020), which searched the number of subclasses for every superclass using a quality metric."
>
> Q18. Figure 1 --> Appears to be a typo in the caption, all of the subclass means have the same subscript (“1”)
>
> Thanks for pointing it out. We revised it.
>
> Q19. "... and tune $\sigma$ to adjust the relativity between super- and sub-classes" What range does this usually fall in? Worth reporting this.
>
> First, please let us clarify a typo. In our experiments, we tuned the value of $\sigma^2$ instead of $\sigma$ because $\sigma^2$ resemble the temperature scaling as discussed in the paragraph below Eq. (7). We tested $\sigma^2$ with values in {0.05, 0.1, 0.15, 0.2, 0.25, 0.3}, and observed $\sigma^2$ from 0.1 to 0.2 is generally better than other choices. According to the suggestion, we revised the Appendix and reported the results of this test with some discussions in Appendix D.3, and referred to it in Section 4.2 ("Performance analysis").
>
> Q20. "transport polytope" Recommend providing a brief description/background on this in the Appendix.
>
> Thanks for this suggestion. In the revised draft, we provided a description of "transport polytope" and its relationship with the optimal transport problem in Appendix A.3, and referred to it in the text of Section 3.2 ("E-step"). In the meantime, we become aware of a typo in Section 3.2 ("E-step"), where $Q_{z_{i},i}=q(z_{i}|v_{i}, y_{i})$ should be $Q_{z_{i},i}=q(z_{i}|v_{i}, y_{i})\frac{1}{n}$. Correspondingly, the first $\frac{1}{n}$ in Eq. (6) was removed, and Eq. (14) was corrected (with a constant term which can be omitted during optimization). This correction of typos does not change our solution and algorithm, which are the same as before.
>
> Q21. "It is noteworthy ..." This is an interesting set of observations. Nit: typo in "contrasitive".
>
> Thanks for pointing it out. We revised the typo.

---

> ### Author Response · Authors · 2021-11-16
> **Response to the comments on Experiments (Part 1)**
>
> Q22. "visual granularity" What exactly does this mean? A comment would be helpful.
>
> Here, it means the granularity of the class hierarchy by watching the images (without referring to their labels). For example, images of class "dog" and class "cat" are of similar visual granularity (and at the same class hierarchy), and images of class "pug" and "beagle" (different breads of dog) are of another similar visual granularity (and at a more fine-grained class hierarchy).
>
> Q23. "20/6/8 splits" So the superclasses are disjoint in train/test/val?
>
> Yes. This dataset is to evaluate the case when there are superclasses not used for model pre-training, but their subclasses may appear in the support set of the testing tasks. These superclasses are what we described as "unseen superclasses" before. This evaluation case is more challenging than evaluating subclasses belonging to the superclasses that were used for pre-training. The evaluation setup follows (Bukchin et al., 2021). We think both cases are practical and meaningful, and worth study.
>
> Q24. "which adds an embedder" What exactly does this embedder do/why is it added?
>
> Adding an embedder follows the practice of some existing works such MoCo-v2 (Chen, et al. 2020b) and SimCLR (Chen, et al. 2020a), which was also used in the different compared methods in (Bukchin et al., 2021). We found there is a lack of discussions in these papers on how could the added embedder help improve performance, but they empirically found adding certain types of the embedder (e.g., MLP) is helpful. We also have the same observation in our experiments. We think the effectiveness may come from (1) more capacity, and (2) better generalization in some cases (if the embedder is dropped at the testing stage, the representation learned by the backbone network may be more generalizable, because the embedder may be specific to the pre-training tasks and drop it may help alleviate overfitting to some extent). For comprehensiveness, we evaluated both of the cases when an embedder is added or not in our experiments, as described in Section 4.1 ("Baselines" and "Implementation").
>
> Q25. "dim" --> "dimension"
>
> We revised it.
>
> Q26. Table 1. Please provide detail on how confidence intervals were computed, and what statistical approach was used to determine superiority.
>
> Following (Tian et al., 2020), for each method, we reported mean accuracy of 1000 random episodes for each case (i.e., 5-way, all-way, etc.) during testing. The 95% confidence interval of mean based on t-distribution was calculated using the accuracy values of the 1000 random episodes. We revised the first paragraph of Section 4.2 to make the description more clear. To determine significance, two-sample t-test can be used. Previously, we didn't thoroughly perform the tests between each baseline method and our proposed method, and considered significant difference if the confidence intervals of two methods do not overlap (a significant difference is also possible if the confidence intervals overlap, which requires statistic tests). Therefore, we revised the sentence "SCGM significantly outperforms ANCOR(-fc) in most cases" to "SCGM significantly outperforms ANCOR(-fc) in most all-way cases" to make it more precise.
>
> Specifically,
> * SCGM-A vs ANCOR: 3 out of 4 all-way cases (Living17, Nonliving26, Entity30)
> * SCGM-A vs ANCOR-fc: 4 out of 4 all-way cases (Living17, Nonliving26, Entity13, Entity30)
> * SCGM-G vs ANCOR: 3 out of 4 all-way cases (Living17, Nonliving26, Entity30)
> * SCGM-G vs ANCOR-fc: 3 out of 4 all-way cases (Living17, Nonliving26, Entity13)
>
> Regarding the discussion on the 5-way cases, please see the response to next question.
>
> Q27. Table 1. Seems to be overlapping confidence intervals in a couple of cases between SCGM and ANCOR.
>
> We observed the cases where the confidence intervals between SCGM and ANCOR overlap, especially on 5-way cases. 5-way cases have less room to improve than the all-way cases, because the randomly selected subclasses are more likely to be from different superclasses in the 5-way cases, which may degrade the evaluation to regular classification of superclasses that cannot clearly show how subclasses were learned. Therefore, we investigated an "intra-class" case in Table 3, where all subclasses of a random superclass were sampled in each episode. From Table 3, we observed less overlapped confidence intervals between SCGM and ANCOR, which may help indicate the superiority of SCGM in distinguishing subclasses. Also, compared to ANCOR, SCGM is more efficient, as demonstrated in Table 4. We described these in Section 4.2 ("Evaluation case (1)"). Here, we chose to evaluate the 5-way cases (rather than more ways such as 10-way or 20-way) is to make the evaluation setup consistent with the setup in existing works (Bukchin et al., 2021, Tian et al., 2020).

---

> ### Author Response · Authors · 2021-11-16
> **Response to the comments on Experiments (Part 2)**
>
> Q28. Table 1. What's similar about the cases where the method outperforms and doesn't outperform ANCOR?
>
> The similarity between the cases is they both are n-way k-shot evaluations. Since SCGM outperforms ANCOR in most all-way cases, the difference is whether the evaluation involves more subclasses (that could be from the same superclass). If there are more such subclasses, SCGM tend to outperform ANCOR, which helps demonstrate its effectiveness in learning subclasses. We are not sure whether we understand this question correctly. We appreciate if the reviewer could clarify it if there is any misunderstanding.
>
> Q29. "baselines on contrastive learning cannot be applied ..." I would cut it, or at least acknowledge that this is not a fundamental limitation.
>
> We cut this piece in the draft according to the suggestion. The response to this question is summarized in the response to General Question Q4.

---

> ### Author Response · Authors · 2021-11-16
> **Response to the comments on Appendix A (Part 1)**
>
> Q30. Eq. (10). I am not following exactly how Jensen’s inequality results in the move from line 1 to line 2 here.
>
> In the revised Appendix A.2, we added one step between the original line 1 and line 2 to make it more explicit and consistent with the equations in the link provided by the reviewer. In the added step, $q(z_{i}|v_{i}, y_{i})$ corresponds to $a_{i}$, the fraction after $q(z_{i}|v_{i}, y_{i})$ corresponds to $x_{i}$. Note $\sum_{z_{i}=1}^{r}q(z_{i}|v_{i}, y_{i}) = 1$. Then, because the log function is concave, using the Jensen's inequality, the third step in Eq. (10) can be obtained.
>
> Q31. "the second step in Eq. (11) ..." I think this is a typo.
>
> Yes. We revised it to Eq. (10) in the current Appendix A.4.
>
> Q32. "$\lambda \ge 1$" Am I correct in understanding that there is strict equality for $\lambda = 1$?
>
> If $\lambda = 1$, the left hand side (LHS) and the right hand side (RHS) of Eq. (14) (in the current draft) are equal. If $\lambda > 1$, the LHS is greater than the RHS. Therefore, choosing a $\lambda \ge 1$ ensures the RHS of Eq. (14) is a valid lower bound of the LHS. Since the LHS is a valid lower bound of our likelihood function in Eq. (5) according to Eq. (10), the RHS of Eq. (14) is a valid lower bound of our likelihood function. Thus, optimizing the RHS of Eq. (14) (i.e., the objective function in Eq. (6)) provides a reasonable solution (approximation) to the likelihood function in Eq. (5).
>
> We added these descriptions in Appendix A.4.
>
> Q33. "equal partition constraint" Does this imply that the constrained transport polytope formulation assumes that subclasses are of equal size? This should be further discussed.
>
> The equal partition constraint theoretically assumes subclasses are of equal or similar sizes, by using a vector $Q1_{n}$ = $\frac{1}{r}1_{r}$ in Eq. (6) (i.e., $a=\frac{1}{r}1_{r}$ in Eq. (11)) as a prior distribution of the subclass sizes. It is possible to replace $\frac{1}{r}1_{r}$ by other prior distributions to reflect an uneven distribution of the subclasses sizes. However, since in this work we assume we don't have such prior knowledge (as we assume subclasses are unobserved in the training dataset), we used a uniform prior. This prior is inspired by (Asano et al., 2020). If we don't have such a prior assumption, the searching space of subclasses will be exponential (because it involves a subproblem of data partition similar to the k-means clustering problem, which is computationally hard to solve (https://en.wikipedia.org/wiki/K-means_clustering)). Therefore, to make the optimization feasible, we considered to use this constraint to help reduce the searching space.
>
> We agree with the reviewer that this uniform prior may not always be reasonable in different applications/tasks (e.g., if there are rare subclasses as discussed by (Sohoni et al., 2020)). However, we found it is effective to our cross-granularity adaptation problem since this problem has less emphasis on exact identification of every subclass. Compared to other baseline methods, the performance improvements indicate reasonable boundaries between subclasses can be learned for model adaptation with this constraint, though the boundaries may not be ideal. Although subclasses have the equal partition constraint, superclasses can have different sizes because they can have different numbers of subclasses.
>
> We recently checked the size distribution of subclasses in the training datasets of the benchmark datasets, and observed the subclasses are of similar sizes on Living17, CIFAR100, and of various sizes on Nonliving26 (from 807 to 1203), Entity13 (from 872 to 1193), Entity30 (from 675 to 1203), TieredImageNet (from 732 to 1300). The size does not vary in a big range, and the majority of the sizes are similar. This may contribute to the performance of our proposed model to some extent. However, we didn't choose these datasets specifically for favoring our proposed method. We used them because the same datasets have been used in (Bukchin et al., 2021), and we try to make the evaluation setup consistent. From this perspective, we may miss an evaluation on datasets with largely uneven subclass sizes for whom may concern. This could be a substantial extension (perhaps a different task that requires some specific designs of the method for improving the worst-case subclass detection accuracy) to the current work, which may be sort of out of the scope of the work. According to the reviewer's suggestion, currently we added a discussion on this limitation in Appendix B.1 to make it clear.

---

> ### Author Response · Authors · 2021-11-16
> **Response to the comments on Appendix A (Part 2)**
>
> Q34. Eq. (13). You remove the 1 constant. Why is this true/necessary?
>
> In the current draft, the previous Eq. (13) becomes Eq. (16). Removing constant 1 is because $\exp{(\frac{(\mu_{z_{i}}^{\top}\cdot\mu_{y_{i}} - 1)}{\sigma^{2}})} =  \frac{\exp{(\mu_{z_{i}}^{\top}\cdot\mu_{y_{i}} / \sigma^{2})}}{\exp{(1 / \sigma^{2})}}$, and the term corresponding to 1, i.e., $\exp{(1 / \sigma^{2})}$, will appear in both of the numerator and denominator of the first term in the bracket in Eq. (15), which can be canceled out. We wrote it in the previous way is for being consistent with Eq. (7), but it seems lead to some confusion. We revised both Eq. (16) and Eq. (17) by keeping the constants, and described that the constants are canceled out in Eq. (15) in the following text.

---

> ### Author Response · Authors · 2021-11-16
> **Response to the comments on Appendix B**
>
> Q35. "... robust learning rate ..." Could be worth searching over robust learning rate for this baseline.
>
> Yes, in the previous experiments, we searched this robust learning rate for this baseline within {0.01, 0.001, 0.0001}, where 0.01 seems to be a default number in the original code. In our experiments, we found 0.0001 is generally better than the other choices.

---

> ### Author Response · Authors · 2021-11-16
> **Response to the comments on Appendix C**
>
> Q36. Table 8. Suggest adding bolding to this table.
>
> We added boldface in the revised draft. Previously, we didn't use boldface in this table because all methods become to have similar performance when there are 20 shots. This is because the 5-way task is close to the regular superclass classification (as discussed in the response to Q27), and the performance limit can be quickly reached by adding more shots. This can be seen from Coarse+'s results, which become close to other models when there are 20 shots in the support set. We described these in Appendix D.1.

---

### Official Review · Reviewer_s1rd · 2021-11-03

**Correctness:** 4
**Technical Novelty And Significance:** 3
**Empirical Novelty And Significance:** 4
**Recommendation:** 8
**Confidence:** 4

**Main Review:**

Pros:
- paper is clearly written and well motivated
- the approach is novel
- empirical results are convincing and very solid
- authors provide code and instructions to recreate datasets for reproducibility, datasets are public

Cons:
- Please provide bullet point list of contributions. I advise to replace the last paragraph in the intro, which is too wordy and is not to the point
- There have been some work studying the relationship between coarse and fine-grained level supervision in the zero-shot classification domain, for example https://arxiv.org/pdf/1906.11892.pdf. Please discuss how your work is different.
- Additional experiments on well established fine-grain classification datasets such as CUB and Flowers would significantly strengthen the results of the study. I beleive those datasets are much more relevant to the task than CIFAR100, for example.
- Testing only on seen / unseen superclasses is limited in that is overlooks the problems that arise in the practical setting when both seen and unseen classes are expected to be mixed. This has been pointed out in relation to generalized zero-shot learning https://arxiv.org/pdf/1712.00981.pdf.

**Summary Of The Paper:**

The paper addresses the problem of transferring knowledge from abundant data with low granularity labels to the fine grained data with few labels. The model uses gaussian mixture approach to link the structures in the low granularity data with the few fine grained labels.

**Summary Of The Review:**

I recommend accept since there is clear novelty in the paper, problem is important and well motivated, empirical results are convincing

---

> ### Author Response · Authors · 2021-11-16
> **Response to the comments from the reviewer (Part 1)**
>
> Thank you so much for the constructive feedback. We sincerely appreciate your valuable suggestions and questions. The following are our responses.
>
> Q1. Please provide bullet point list of contributions.
>
> Thanks for the suggestion. In the revised draft, we revised the last paragraph of Section 1 by using bullet point list of contributions.
>
> Q2. There have been some work studying the relationship between coarse and fine-grained level supervision in the zero-shot classification domain, for example https://arxiv.org/pdf/1906.11892.pdf. Please discuss how your work is different.
>
> We read the suggested paper (Oreshkin et al., 2020), and the following is our summary of difference. In (Oreshkin et al., 2020), the authors target the fine-grained visual description scenario, which was defined by (Reed, et al. "Learning deep representations of fine-grained visual descriptions." In CVPR, 2016.). Their problem is to classify images with subtle distinctions (Reed, et al., 2016), by leveraging the auxiliary information in textual description of the images (which is what they call fine-grained visual description), in the zero-shot learning scenario. Their method is designed to embed both images and texts jointly, and train the model through optimizing image/text retrieval losses and image/text classification losses. Therefore, their problem definition is different from our cross-granularity few-shot (CGFS) learning problem, and their method is not designed to learn fine-grained embeddings (that distinguishes different subclasses) by only using the coarse labels (of superclasses) for training. In their problem definition, they didn't describe the class hierarchy, and didn't distinguish superclass and subclass. Thus, the classes of the testing images may be of the same level of taxonomy as their training classes, i.e., the training and testing scenarios do not "cross-granularity". In contrast, we focus on training a model using the coarse labels (of superclasses), and the pre-trained model is to be adapted to a testing task using few samples with fine-grained labels (of subclasses). The model is then evaluated on the query set of that task where the samples belong to the subclasses. Thus, the training and testing scenarios "cross-granularity". Accordingly, our proposed model is remarkably different from the method in (Oreshkin et al., 2020) because of the different goals. Our proposed model is designed for learning embeddings that distinguish subclasses by only using superclass labels, which facilitate the few-shot cross-granularity adaptation.
>
> Q3. Additional experiments on well established fine-grain classification datasets such as CUB and Flowers.
>
> We investigated the suggested CUB and Flowers datasets, which we found were used by (Oreshkin et al., 2020) and (Reed, et al., 2016). These datasets contain some attributes and rich fine-grained textual descriptions of images with subtle distinctions. However, as far as we understand, they don't provide hierarchy of classes, i.e., superclasses and subclasses, nor information about how subclasses associate with superclass. Because in the CGFS task, we try to train the compared models with superclass labels, and evaluate their performance on the subclasses, the information about the class hierarchy is indispensable for the evaluation (i.e., for determining the subclasses of the support set, for computing the accuracy on the subclasses of the query set, etc.). Therefore, we think the CUB and FLowers datasets may not be suitable for our experiments. In our experiments, we used the 6 benchmark datasets as summarized in the first table in Section 4 because they provided the class hierarchy information, and were used in other works, such as (Bukchin et al., 2021). So we try to make the evaluation setup consistent with the existing work.

---

> > ### Comment · Reviewer_s1rd · 2021-11-16
> > **Reviewer s1rd, post rebuttal response**
> >
> > I would like to sincerely thank the authors for their detailed responses. I have reviewed the responses and it appears to me that the authors have addressed all of the concerns posed by reviewers. Therefore, I retain my acceptance recommendation.

---

> > > ### Author Response · Authors · 2021-11-17
> > > **Thank you**
> > >
> > > Thank you for your recognition of our work!

---

> ### Author Response · Authors · 2021-11-16
> **Response to the comments from the reviewer (Part 2)**
>
> Q4. Testing only on seen / unseen superclasses is limited in that it overlooks the problems that arise in the practical setting when both seen and unseen classes are expected to be mixed. This has been pointed out in relation to generalized zero-shot learning https://arxiv.org/pdf/1712.00981.pdf.
>
> Thanks for this insightful comment. We read the suggested paper (Xian et al., 2018), and found its investigated problem is almost the same as (Oreshkin et al., 2020). So its problem is different from our work for the same reason as described in the response to Q2. We agree with the reviewer that the case of mixed seen and unseen classes is practical and worth more discussions, because there could be an imbalance of prediction performance between seen and unseen classes due to the inductive bias. Although in our work, both the subclasses of seen superclasses and the subclasses of unseen superclasses are disjoint with the seen superclasses in the training set (because all subclasses and all superclasses are two disjoint sets), it may be possible that prediction is more towards the former than the latter. This imbalance may also lead to misclassification of the subclasses of unseen superclasses, to those of seen superclasses. Thus it could be an open challenge in the CGFS task which may be rarely discussed in other works, and it may require some specific technical designs to alleviate the imbalance. Currently, we used the two fundamental cases without mixing subclasses of seen and unseen superclasses to facilitate understanding of our task (similar to (Bukchin et al., 2021)), and we added a discussion about the more complicated case in Appendix B.2 and cited the suggested papers in the revised draft.

---

### Author Response · Authors · 2021-11-16
**Appreciate your attention and time**

Dear Reviewers,

We sincerely appreciate your valuable comments that help us refine our work. If you have more questions or concerns about our response or current draft, please let us know. We are happy to discuss with you.

---

### Decision · Program_Chairs · 2022-01-20

**Decision:**

Accept (Spotlight)

**Comment:**

This work presents an approach to learning good representations for few-shot learning when supervision is provided at the super-class level and is otherwise missing at the sub-class level.

After some discussion with the authors, all reviewers are supportive of this work being accepted. Two reviewers were even supportive of this work being presented at least as a spotlight.

The approach presented is well motivated, experiments demonstrate its value and include a nice application in the medical domain, making the work stand out relatively to most work in few-shot classification. Therefore, I'm happy to recommend this work be accepted and receive a spotlight presentation.